# M5-brane prongs, string soliton bound states and wall-crossing

Varun Gupta, K. Narayan

*Chennai Mathematical Institute,*
*SIPCOT IT Park, Siruseri 603103, India.*

## Abstract

We study abelian M5-brane field configurations representing BPS bound states of self-dual string solitons whose locations correspond to the endlines of M2-branes ending on the M5-branes. The BPS equations are obtained from appropriate Bogomolny completion of the effective abelian low energy functional with two transverse scalars, using two vectors representing the directions along which these endline strings extend. Then we impose boundary conditions on the scalars near the string soliton cores. This leads to a molecule-like equilibrium structure of two non-parallel string solitons at fixed transverse separations, with the M5-brane "prong" deformations comprising two "spikes", each shaped like a ridge. The resulting picture becomes increasingly accurate as one approaches the wall of marginal stability, on which these states decay. There are various parallels with wall-crossing phenomena for string web configurations obtained from D3-brane deformations.

# 1  Introduction

M5-branes are fascinating objects in M-theory. The theory on a single M5-brane in flat space is free, comprising a self-dual anti-symmetric 3-form tensor field strength and five scalar fields and fermions. A stack of coincident M5-branes is described by an interacting non-abelian 3-form theory with massless scalars. Since the self-dual 3-form field strength arises from a 2-form potential which couples to a string-like object, the M5-brane theory appears to be a theory of interacting self-dual strings: these strings can be thought of as the endlines of M2-branes that end on the M5-brane. A succinct description of various aspects appears in [1] and some early nice papers include *e.g.* [2, 3, 4, 5]. While we do not understand this theory fully, we do know several aspects from various points of view: a classic old review is [6]. More recent reviews of several aspects include *e.g.* [7, 8]. The extensive investigations of holography [9, 10, 11] over the last several years suggest that the M5-brane $(2, 0)$ theory can be aptly regarded as an abstract conformal field theory (see *e.g.* [12] for a bootstrap formulation) and the self-dual strings above are then solitonic objects.

When the M5-branes in a stack are separated, the M2-branes stretched between them lead to charged massive states in the M5-theory "Higgsed", loosely speaking (more precisely, the M5-theory on the tensor branch). A single M2-brane stretched between two parallel separated M5-branes ends on a line in the M5-worldvolume thus giving rise to a half-BPS self-dual string soliton state in the worldvolume M5-brane theory [13, 14]. As a field configuration in the full "Higgsed" M5-brane nonabelian tensor theory, this is perhaps a line-charge analog

of the BPS 't Hooft-Polyakov monopole in super Yang-Mills theory (see *e.g.* [15] for a review), and can be described at least in the low energy abelian "Higgsed" theory in a simple way (as we will review below). Consider the theory of three parallel M5-branes: two non-parallel M2-branes stretching between one M5 and the other two energetically lead to a nontrivial M5-M2-M2 bound state. These sorts of states can be recognized readily in a different regime. They simply correspond to nontrivial string webs stretching between three parallel separated D3-branes in the IIB string theory obtained by compactifying M-theory on $T^2$: see [16], where these M2-brane webs in M-theory were in fact already anticipated. The M5-branes become D3-branes while the M2-branes wrapping appropriate $T^2$-cycles become various $(p, q)$-strings (as reviewed in *e.g.* [17]). The line charges from the M2-branes ending on the M5-brane compactify to point charges in the Higgsed gauge theory on the D3-branes.

These string webs can in fact be obtained as nontrivial brane deformations in the D3-brane worldvolume super Yang-Mills theory. As a BPS state, a single string ending on a single D3-brane is described in the low energy abelian theory by the BPS bound equation $\vec{E} = \nabla X$ with $\vec{E}$ the electric field and $X$ one of the transverse scalars. The resulting field configuration $X \sim \frac{q_e}{|\vec{r}-\vec{r}_0|} - X_0$ can be interpreted as a "brane spike" worldvolume deformation emanating from the D3-brane at the location of the charge [18]. Likewise string web states can also be described as more general "brane prongs" in the low energy abelian theory of three or more Higgsed D3-branes [19, 20, 21]: this low energy description was developed from several previous investigations of such states including [22] as well as *e.g.* [24]-[34] (see also the review [35]). These are nontrivial molecule-like dyon bound states in the Higgsed vacua of the $\mathcal{N}{=}4$ $U(N)$ super Yang-Mills theory on the D3-branes: they decay across walls of marginal stability and the wall-crossing phenomena are in fact well-described by these brane prong field configurations in the low energy abelian theory (the qualitative picture has parallels with similar observations in BPS black hole bound states [36]).

Motivated by these investigations, in this paper, we consider abelian M5-brane field configurations representing BPS bound states of self-dual string solitons whose locations correspond to the endlines of M2-branes ending on the M5-branes. A single M2-brane ending on a single M5-brane leads to a $\frac{1}{2}$-BPS state described by a ridge-shaped *"spike"* deformation of the M5-brane with one transverse scalar excited as $X \sim \frac{q}{|\vec{\rho}-\vec{\rho}_0|^2} - X_0$. This M5-brane "spike" field configuration characterizes the self-dual string that is the endline of the M2-brane ending on the M5-brane and was discussed in [13, 14] (reviewed in [6]). Appropriate combinations of multiple M2-branes give $\frac{1}{4}$-BPS states described by M5-brane *"prong"* deformations: the structure of these is more intricate. They can be thought of as a superposition of two spikes corresponding to a bound state of two non-parallel string solitons at fixed separation transverse to their extended directions, which dovetails with nontrivial boundary conditions on the scalar moduli in the vicinity of the string soliton

"cores". Under $M \to IIB$ duality obtained from compactifying M-theory on $T^2$, they are expected to reduce to D3-brane prongs representing string webs stretched between multiple parallel non-coincident D3-branes described above.

The BPS bound equations in the M5-brane are obtained by performing a Bogomolny completion of the M5-brane energy functional comprising a free self-dual 3-form field strength $H_{abc}$ and two transverse scalars. This was already set up in a preliminary manner in [22]: here we will develop this more elaborately. It is convenient to recast $H_{abc}$ in terms of a spatially "dual" 2-form object $\tilde{H}_{ab} \equiv \star_5 H$ (where $a, b$ are 5-space indices alone, reflecting the self-duality of $H$): this is then Bogomolny-squared with the scalar derivatives using two vectors $\zeta^a$ to obtain BPS bound equations (in one form, these were obtained from supersymmetry in [23]; our analysis, complementing that, is more fuelled by wall-crossing considerations and the D3-brane string web descriptions). These $\zeta$-vectors have support only in a 2-dim (4-5) subspace of the 5-dim worldvolume of the M5-brane: this simulates the intuition that upon compactifying this 2-subspace as $T^2$ the line charges wrap the $\zeta$-directions becoming point charges in the noncompact 3-dimensions. In this light, these equations can be decomposed into components along the $4-$ and/or $5-$ directions, and some of these components then resemble the BPS equations for the D3-brane. Another crucial ingredient in the M5-description is the fact that the scalar field configurations are subject to "line source conditions" encoding translation invariance along the string, reflected in the ridge-shape. These alongwith the BPS bound equations and the moduli boundary conditions lead us to the spike superpositions in the prong field configurations.

The moduli boundary conditions in the vicinity of the string cores we impose are Dirichlet boundary conditions on the scalars at the closest point of approach of the two strings: these imply fixed separations transverse to the extended directions of the strings. For generic finite separation, these lead to nontrivial variations in the moduli as we go far along the strings, which amounts to nontrivial M5-brane bendings. However in the limit of approaching the wall of marginal stability, we show that these variations become negligible and the moduli values become near constant all along the strings. Thus our formulation reveals self-consistent approximate configurations which become increasingly accurate in the marginal stability limit, as we will describe elaborately.

In sec. 2, we describe the Bogomolny completion in the M5-brane low energy abelian theory, while sec. 3 describes the simplest M5-brane prong configurations and the corresponding string soliton bound states with the effective 1-string tension and the wall-crossing limit in sec. 3.3 (with some further comments in sec. 3.4). More general states appear in sec. 4. In sec. 5 we discuss embedding these field configurations in multiple abelian "Higgsed" M5-brane theories. A Discussion comprising an overview and related questions appears in 6, and some technical details in Appendices.

# 2  Bogomolny completion in M5 effective field theory

To describe the M5-brane spatial worldvolume for the field configurations and states to follow, we will find it useful to define the notation

$$\vec{\rho} = (x_1, x_2, x_3, x_4, x_5) \equiv (\vec{r}, \vec{x}) \; ; \qquad \rho_X \equiv (\vec{r}, x_5) \; ; \qquad \rho_Y \equiv (\vec{r}, x_4) \; ,$$
$$\rho_\zeta \equiv (\vec{r}, \vec{x}), \quad \text{with} \quad \vec{x} \cdot \zeta = 0 \; . \tag{2.1}$$

*i.e.* $\vec{\rho}$ is the coordinate describing the 5-dimensional spatial worldvolume of the M5-brane which decomposes into a 3-dim subspace spanned by $\{x_1, x_2, x_3\}$ and the remaining 2-dim subspace which $\{x_4, x_5\}$ spans. $\vec{r} \equiv (x_1, x_2, x_3)$ denotes the coordinate in the 3-dim subspace. $\vec{\rho}_X$ denotes the coordinate in the 4-dimensional subspace orthogonal to the $x_4$-direction and likewise $\vec{\rho}_Y$ the coordinate in the 4-dim subspace orthogonal to the $x_5$-direction. In the last line, we have chosen a general vector $\zeta \in (x_4, x_5)$ in the 45-plane and denoted the 4-dim subspace orthogonal to $\zeta$ by the coordinates $\vec{\rho}_\zeta$.

Now we introduce $\zeta^1$, $\zeta^2 \in (x_4, x_5)$ as two unit vectors in the 45-subspace in the 5-dim worldvolume, and define (with $a, .., e$ only spatial indices $\{1, 2, 3, 4, 5\}$)

$$\zeta^1, \zeta^2 \in (x_4, x_5) : \qquad \tilde{\mathcal{H}}^{ab} = \frac{1}{6} \epsilon^{abcde} H_{cde} \; , \qquad |\tilde{\mathcal{H}}|^2 = \tilde{\mathcal{H}}^{ab} \tilde{\mathcal{H}}^{cd} \delta_{ac} \delta_{bd} \; ,$$
$$|\partial X|^2 = \delta_{ab} \partial_a X \partial_b X \; , \qquad \partial_{[a} X \zeta^1_{b]} = \frac{1}{2} \left( \partial_a X \zeta^1_b - \partial_b X \zeta^1_a \right) \; . \tag{2.2}$$

Since these are all spatial indices with a flat metric, we will not distinguish up/down indices.

The presence of precisely two $\zeta$-vectors dovetails with including only two of the five transverse scalars to the M5-brane. This corresponds to including only two M2-branes ending on the M5-brane: in an essential sense, this is incomplete from the point of view of M5-M2 systems. However restricting to this, which reflects the decomposition (2.1), is adequate from the point of view of uplifts of D3-brane configurations describing string webs. In other words, this restriction implicitly encodes this subspace of $M \to IIB$ dualities obtained from compactifications of M-theory on $T^2 \equiv (x_4, x_5)$. With this in mind, the energy functional for static configurations in a single free M5-brane theory comprising the self-dual 3-form field strength and two transverse scalars is

$$\mathcal{E} = \frac{1}{2} \left( |\partial X|^2 + |\partial Y|^2 + \frac{1}{2} |\tilde{\mathcal{H}}|^2 \right). \tag{2.3}$$

The sum here is over only spatial indices in $\tilde{H}$: this incorporates the fact that $H$ is a self-dual 3-form field strength with $H = \star_6 H$. This can be abstracted from a superbrane formulation as discussed in [6] (see also *e.g.* [37, 14, 22]). We will find it adequate to focus on the minimal low energy point of view of a single free M5-brane and its field content here,

obtained by retaining only the lowest mass dimension terms. The action for an M5-brane with tension $T_6 \sim \frac{1}{l_{11}^6}$ (with $l_{11}$ the 11-dim Planck length) [17] is written in terms of the M5-brane coordinates $x^I$ in transverse space: these are redefined to M5-brane worldvolume field theory scalars as

$$\frac{1}{l_{11}^6} \int d^6s \, (\partial x^I)^2 \; \equiv \; \int d^6s \, (\partial X^I)^2 \; ; \qquad X^I \equiv \frac{1}{l_{11}^3} \, x^I \; , \qquad (2.4)$$

so the transverse scalar fields $X^I \equiv X, Y, \ldots$ are manifestly dimension-2, and are the relevant observables in the field theory regime. The 2-form $B_{\mu\nu}$ has mass dimension-2 (so the Wilson surface operator $\int B$ is dimensionless) and the scalars have mass dimension-2, so all terms in $\mathcal{E}$ are dimension-6.

Upon Bogomolny completion of squares using the above $\zeta$-decomposition, as in fact anticipated in [22] (see Appendix A for some useful details), this gives

$$\mathcal{E} = \left| \partial_{[a} X \zeta_{b]}^1 + \partial_{[a} Y \zeta_{b]}^2 - \frac{1}{2} \delta_{ac} \delta_{bd} \tilde{\mathcal{H}}^{cd} \right|^2 + \frac{1}{2} \left( \zeta_a^1 \partial_a X + \zeta_a^2 \partial_a Y \right)^2 \qquad (2.5)$$

$$+ \partial_a X \tilde{\mathcal{H}}^{ab} \zeta_b^1 + \partial_a Y \tilde{\mathcal{H}}^{ab} \zeta_b^2 - \sum_{a \neq b} \zeta_a^1 \zeta_b^2 \left( \partial_a X \partial_b Y - \partial_b X \partial_a Y \right) - \left( \zeta^1 \cdot \zeta^2 \right) \left( \partial X \cdot \partial Y \right)$$

Then the energy functional is minimized when the BPS bound equations

$$\partial_{[a} X \zeta_{b]}^1 + \partial_{[a} Y \zeta_{b]}^2 - \frac{1}{2} \delta_{ac} \delta_{bd} \tilde{\mathcal{H}}^{cd} = 0 \; ,$$

$$\zeta_a^1 \partial_a X + \zeta_a^2 \partial_a Y = 0 \; , \qquad (2.6)$$

are satisfied. We will study these equations and the resulting configurations alongwith appropriate boundary conditions in what follows. The above essentially ignores the last two sets of terms in (2.5): we will argue later in sec. 3.3, sec. 3.4 and sec. 4.3 that these terms in fact give negligible contribution in the limit of approaching the wall of marginal stability, and thereby lead to a self-consistent energy minimization via the Bogomolny completion.

The 3-form field strength and its equation of motion can be recast giving

$$H_{abc} = \epsilon_{abcde} \tilde{\mathcal{H}}^{de} \quad \rightarrow \quad d \star H = 0 \quad \rightarrow \quad \partial_a \tilde{\mathcal{H}}^{ab} = 0 \; . \qquad (2.7)$$

Thus

$$\sum_{\substack{a \\ a \neq b}} \partial_a (\partial_a X \zeta_b^1 - \partial_b X \zeta_a^1) + \sum_{\substack{a \\ a \neq b}} \partial_a (\partial_a Y \zeta_b^2 - \partial_b Y \zeta_a^2) = 0 \; . \qquad (2.8)$$

Imposing line source conditions along $\zeta^1, \zeta^2$ directions leads to harmonicity in 4-dim subspaces orthogonal to the $\zeta$-vectors, and thereby to harmonic field configurations transverse to string soliton line sources stretched along $\zeta^1, \zeta^2$, roughly "superposing" the two string solitons as we will describe.

It turns out that these are best described in component form, which we will now proceed to do. In component form ($a = 1, 2, 3$), we define

$$E^a = \tilde{\mathcal{H}}^{a4} , \qquad B^a = \tilde{\mathcal{H}}^{a5} , \qquad \Pi = \tilde{\mathcal{H}}^{45} ; \qquad \vec{E} = E^a \hat{x}^a , \quad \vec{B} = B^a \hat{x}^a . \qquad (2.9)$$

It is consistent to set $K^a = \frac{1}{2}\epsilon^{abc}\tilde{\mathcal{H}}^{bc} = 0$. In terms of the 3-form $H$, these components are $B^a = \frac{1}{2}\epsilon^{abc}H_{bc4}$, $E^a = \frac{1}{2}\epsilon^{abc}H_{bc5}$, $\Pi = H_{123}$, $K^a = H_{a45}$.

Taking $\zeta^1$, $\zeta^2$ to lie in the $\{x_4, x_5\}$-plane (*i.e.* no components in $\{1, 2, 3\}$), the BPS equations (2.6) in component form become

$$E_a = \partial_a(X\zeta^1_4 + Y\zeta^2_4) , \qquad B_a = \partial_a(X\zeta^1_5 + Y\zeta^2_5) , \qquad \Pi = \partial_4(X\zeta^1_5 + Y\zeta^2_5) - \partial_5(X\zeta^1_4 + Y\zeta^2_4) ,$$

$$(\zeta^1_4\partial_4 + \zeta^1_5\partial_5)X + (\zeta^1_4\partial_4 + \zeta^1_5\partial_5)Y = \partial_4(X\zeta^1_4 + Y\zeta^2_4) + \partial_5(X\zeta^1_5 + Y\zeta^2_5) = 0 . \qquad (2.10)$$

Defining appropriate $X', Y'$, leads to somewhat simpler expressions:

$$X' \equiv X\zeta^1_4 + Y\zeta^2_4 , \qquad Y' \equiv X\zeta^1_5 + Y\zeta^2_5 ,$$

$$\vec{E} = \nabla X' , \qquad \vec{B} = \nabla Y' , \qquad \Pi = \partial_4 Y' - \partial_5 X' , \qquad \partial_4 X' + \partial_5 Y' = 0 . \qquad (2.11)$$

The equations in this form were also obtained via supersymmetry in [23]. Our analysis of the resulting field configurations is motivated by wall-crossing phenomena and string webs via D3-brane prongs: as we will see, the map between $X', Y'$ and $X, Y$, involving arbitrary vectors $\zeta^{1,2}$ in the $(x^4, x^5)$-plane allows for constructing string-string bound states for arbitrary charges. The Bogomolny completion (2.5) can be written as

$$\mathcal{E} = \frac{1}{2}\left|\vec{E} - \nabla X'\right|^2 + \frac{1}{2}\left|\vec{B} - \nabla Y'\right|^2 + \vec{E} \cdot \nabla X' + \vec{B} \cdot \nabla Y'$$

$$+ \frac{1}{2}(\zeta^1_a\partial_a X + \zeta^2_a\partial_a Y)^2 + \frac{1}{2}\left|\Pi + \partial_5 X' - \partial_4 Y'\right|^2 + \Pi(\partial_4 Y' - \partial_5 X')$$

$$- \sum_{a\neq b}\zeta^1_a\zeta^2_b\left(\partial_a X\partial_b Y - \partial_b X\partial_a Y\right) - \left(\zeta^1 \cdot \zeta^2\right)\left(\partial X \cdot \partial Y\right) \qquad (2.12)$$

The simplest case with two orthogonal vectors $\zeta^1 = (1, 0)$, $\zeta^2 = (0, 1)$, gives $X' = X$, $Y' = Y$.

Taking further derivatives of the last condition in (2.11) gives

$$\partial_4^2 X' + \partial_4\partial_5 Y' = 0 , \qquad \partial_4\partial_5 X' + \partial_5^2 Y' = 0 . \qquad (2.13)$$

The equations of motion then give

$$\partial_a\tilde{H}^{a4} = 0 : \qquad \nabla \cdot \vec{E} - \partial_5\Pi = 0 = \nabla_i^2 X' + \partial_4^2 X' + \partial_5^2 X' ,$$

$$\partial_a\tilde{H}^{a5} = 0 : \qquad \nabla \cdot \vec{B} + \partial_4\Pi = 0 = \nabla_i^2 Y' + \partial_4^2 Y' + \partial_5^2 Y' . \qquad (2.14)$$

In other words, $X', Y'$ are harmonic in the 5-dim spatial M5-worldvolume. Since $X, Y$ are linear combinations of $X', Y'$, this implies that $X, Y$ are also similarly harmonic.

We have in mind two M2-branes ending on the M5-brane: each M2-brane endline defines a self-dual string soliton in the M5-brane. So we have two string solitons stretching along the $\zeta^1$ and $\zeta^2$ directions, which we restrict to lie in the $\{x_4, x_5\}$ plane but take as general otherwise. This suggests that the two scalar field configurations enjoy translation invariance along $\zeta^1$ and $\zeta^2$ directions respectively. In the M5-brane effective field theory this implies that the transverse scalars obey "line source" conditions. One might imagine imposing these line source conditions on $X', Y'$, but for various technical reasons, it turns out more natural to impose

$$\zeta_a^1 \partial_a X = 0 \ , \qquad \zeta_a^2 \partial_a Y = 0 \ , \tag{2.15}$$

on $X, Y$. Thus

$$\nabla_i^2 X + (\zeta_m^{1\perp} \partial_m)^2 X = 0 \ , \qquad \nabla_i^2 Y + (\zeta_m^{2\perp} \partial_m)^2 Y = 0 \ , \tag{2.16}$$

i.e. $X, Y$ are harmonic in 4-dim subspaces transverse to $\zeta^1$ and $\zeta^2$ respectively. This follows from the 5-space harmonicity of $X, Y$ upon imposing the line source conditions. So

$$X = \frac{c_X}{|\vec{\rho}_{\zeta^1} - \vec{\rho}_{\zeta^1}^0|^2} - X_0, \qquad Y = \frac{c_Y}{|\vec{\rho}_{\zeta^2} - \vec{\rho}_{\zeta^2}^0|^2} - Y_0, \tag{2.17}$$

using the notation (2.1), with $c_X, c_Y$ being arbitrary coefficients here. We then use (2.11) to write the field configurations/states. Then (2.11) shows that the $X', Y'$ scalars being arbitrary linear combinations of the "basis" fields $X, Y$ (2.17) give spike legs at general angles.

We take the two strings to be located at $\vec{r} = \vec{r}_1$ and $\vec{r} = \vec{r}_2$, i.e. separated in the 3-space directions, and stretched along lines in the $(x^4, x^5)$-plane in the $\zeta^1$ and $\zeta^2$ directions. By using overall translation invariance along the $x^4, x^5$-directions, we can always orient the strings to pass through the origin $(x^4, x^5) = (0, 0)$. Thus, using (2.1) we can parametrize the coordinates of the two strings as

$$\vec{\rho}_1 = (\vec{r}, x^4, x^5)_1 = (\vec{r}_1, x_1 \zeta_1^4, x_1 \zeta_1^5), \qquad \vec{\rho}_2 = (\vec{r}, x^4, x^5)_2 = (\vec{r}_2, x_2 \zeta_2^4, x_2 \zeta_2^5), \qquad \forall \ x_1, x_2. \tag{2.18}$$

Thus the closest point of approach is at $(x^4, x^5) = (0, 0)$, the separation being $|\vec{r}_1 - \vec{r}_2|$ (see Figure 1 (c) for a specific case discussed later). Now we impose boundary conditions mapping the worldvolume core locations and moduli space: it might seem natural to impose moduli boundary conditions all along the string spatial extents, i.e. for $\vec{\rho}_{\zeta^1} \to \vec{\rho}_{\zeta^1}^0$ and $\vec{\rho}_{\zeta^2} \to \vec{\rho}_{\zeta^2}^0$, but this turns out to be overconstraining at generic points on the M5-vacuum moduli space. Instead we impose only the weaker boundary conditions below, at the closest

approach between the two string solitons:

$$\vec{\rho} \to (\vec{r}_1, 0, 0) : \quad (X, Y) \to (X_1, 0) \quad \Rightarrow \quad (X', Y') \to (X_1 \zeta_4^1, X_1 \zeta_5^1) \ ,$$

$$\vec{\rho} \to (\vec{r}_2, 0, 0) : \quad (X, Y) \to (0, Y_1) \quad \Rightarrow \quad (X', Y') \to (Y_1 \zeta_4^2, Y_1 \zeta_5^2) \ . \qquad (2.19)$$

In what follows, we will analyse the consequences of these boundary conditions on the field configurations and the resulting M5-brane bending geometry.

The boundary conditions suggest that $X', Y'$ lead to spike legs at angles defined by $\zeta^1$ and $\zeta^2$ which being general imply that general prong shapes are captured by the field configurations (2.11) in terms of $X', Y'$. Thus we take $X', Y'$ as the generic transverse scalars to the M5-brane. We will now illustrate this with several examples. The simplest case $(1, 0), (0, 1)$ has $X' = X,\ Y' = Y$, which we will discuss in detail next.

# 3 The simplest M5-M2 BPS states

A simple family of states is obtained when the two M2-branes end on the M5-brane on orthogonal string solitons. Then the $\zeta$-vectors are orthogonal: without loss of generality we can take them to be

$$\zeta^1 = \hat{x}^4 \equiv (1, 0) \ , \qquad \zeta^2 = \hat{x}^5 \equiv (0, 1) \ . \qquad (3.1)$$

Then (2.11) and (2.6) give

$$X' = X \ , \quad Y' = Y \ ; \qquad \vec{E} = \nabla X \ , \qquad \vec{B} = \nabla Y \ , \qquad \Pi = \partial_4 Y - \partial_5 X \ . \qquad (3.2)$$

The line source conditions (2.15) become

$$\zeta^1 \cdot \partial X = \partial_4 X = 0 \ , \qquad \zeta^2 \cdot \partial Y = \partial_5 Y = 0 \ . \qquad (3.3)$$

The 3-form field equations (2.14) alongwith (3.3) now give

$$\nabla_i^2 X + \partial_5^2 X = 0 \ , \qquad \nabla_i^2 Y + \partial_4^2 Y = 0 \ , \qquad (3.4)$$

so the scalars $X, Y$ are harmonic in 4-subspaces orthogonal to $x_4$ and $x_5$ respectively.

## 3.1 A single string soliton: the M5-brane spike

Consider first a single M2-brane ending on the M5-brane giving a self-dual string soliton stretched along say the $x^4$-direction so $\zeta_X = (1, 0)$. The field configuration and boundary condition are

$$X = \frac{q_e}{|\vec{\rho} - \vec{\rho}_0|^2} - X_0 = \frac{q_e}{|\vec{\rho}_X - \vec{\rho}_{X,0}|^2} - X_0 \equiv \frac{q_e}{(\vec{r} - \vec{r}_0)^2 + (x^5)^2} - X_0 \ ,$$

$$\tilde{H}^{a4} \hat{x}^a = \vec{E} = \nabla X \ , \qquad \tilde{H}^{45} = \Pi = -\partial_5 X \ ; \qquad \zeta_X = (1, 0) : \quad \vec{\rho}_X = (\vec{r}, x^5) \ . \qquad (3.5)$$

The first expression for $X$ is written in terms of the 5-dim coordinate $\vec{\rho}$ while the second expression specifies the fact that $X$ is harmonic in the 4-dim subspace transverse to the $x_4$-direction $\zeta_X$ that the string soliton stretches along (the last expression makes this explicit). We see that there is a $\frac{1}{\rho^2}$ divergence in $X(\vec{\rho})$ at the location of the string, parametrized as $\vec{\rho} = (\vec{r}_0, x^4, 0)$ using (2.18): the M2-brane ending on the M5-brane creates an infinitely long spike on the M5-brane at the worldvolume string location where the M2-brane intersects the M5-brane [13, 14]. The fact that the M2-brane extends along the $x_4$ direction means that the "spike" is shaped like a "ridge" with the "flat ridge-top" stretching along $x_4$ in the M5-brane worldvolume, as reviewed in [6]: Figure 1 (a) (or (b)) shows this ridge-shaped M5-brane "spike". We can now impose a boundary condition on the scalar $X$,

$$\vec{\rho} \to \vec{\rho}_{X,0} : \qquad X \to X_1 \ . \tag{3.6}$$

This represents a long distance cutoff in the $X$-space transverse to the M5-brane, which translates to a short distance cutoff around the string soliton core: now the spike is of finite length. Later we will embed these scalar configurations into the theory of multiple M5-branes separated ("Higgsed") and the cutoff will translate to the distance between the M5-branes.

As $\vec{\rho} \to \infty$, we see that $X \to -X_0$: *i.e.* far from the string soliton core, the M5-brane deformation becomes vanishingly small and the scalar $X$ relaxes to its modulus value $-X_0$ characterizing the M5-brane vacuum. Now the boundary condition at $\vec{\rho}_X^0$ above gives

$$q_e \alpha_{00} - X_0 \sim X_1 \qquad \Rightarrow \qquad \frac{1}{\alpha_{00}} \equiv \lim_{\vec{\rho} \to \vec{\rho}_0^+} |\vec{\rho} - \vec{\rho}_0|^2 = (\vec{r}_c - \vec{r}_0)^2 + (x_c^5)^2 \sim \frac{q_e}{X_0 + X_1} \ , \tag{3.7}$$

where $\vec{\rho}_0^+$ means that $\vec{\rho}$ approaches $\vec{\rho}_0$ but is cut off at $\vec{\rho}_c = (\vec{r}_c, x_c^5)$. This leads to a finite "core size": roughly speaking, we obtain a thickened (fuzzy) cylindrical tube. Thus the cross-sectional size, or thickness, of the string core is given by the inverse length of the M5-brane spike in transverse (moduli) space.

For what follows we will find it useful to write the configuration (3.5) more generally as

$$X = \frac{q_e}{|\vec{\rho}_\zeta - \vec{\rho}_{\zeta,0}|^2} - X_0 = \frac{q_e}{(\vec{r} - \vec{r}_0)^2 + (\zeta^\perp \cdot (\vec{x} - \vec{x}_0))^2} - X_0 \ , \qquad \vec{x} \cdot \zeta = 0 \ , \tag{3.8}$$

where $\zeta^\perp \in (x_4, x_5)$ is a unit vector orthogonal to $\zeta$. For instance $\zeta_X = (1, 0)$ has $\zeta_X^\perp = (0, 1)$ recovering the field configuration in (3.5). It is also instructive to note that we can likewise construct similar M5-brane spike configurations carrying generic dyonic charge $(q_e, q_m)$: consider

$$\zeta^1 = \frac{1}{\sqrt{q_e^2 + q_m^2}} (q_e, q_m) \equiv \gamma(q_e, q_m) \ , \qquad \zeta^2 = 0 \ ,$$

$$X' = \frac{q_e}{(\vec{r} - \vec{r}_0)^2 + (\zeta^\perp \cdot (\vec{x} - \vec{x}_0))^2} - X_0' \ , \qquad Y' = \frac{q_m}{(\vec{r} - \vec{r}_0)^2 + (\zeta^\perp \cdot (\vec{x} - \vec{x}_0))^2} - Y_0' \ ,$$

$$\tilde{H}^{a4} \hat{x}^a = \vec{E} = \nabla X' \ , \qquad \tilde{H}^{a5} \hat{x}^a = \vec{B} = \nabla Y' \ , \qquad \tilde{H}^{45} = \Pi = \partial_4 Y' - \partial_5 X' \ . \tag{3.9}$$

We see that $(\vec{E}, \vec{B}) \propto (q_e, q_m)$. We have constructed $X', Y'$ using a single "basis" field $X$ with $Y = 0$: the coefficients in (2.17) are $c_X = 1/\gamma$, $c_Y = 0$. This can be thought of as an electric string "rotated" through a single nontrivial $\zeta^1$-vector to give a dyonic string soliton.

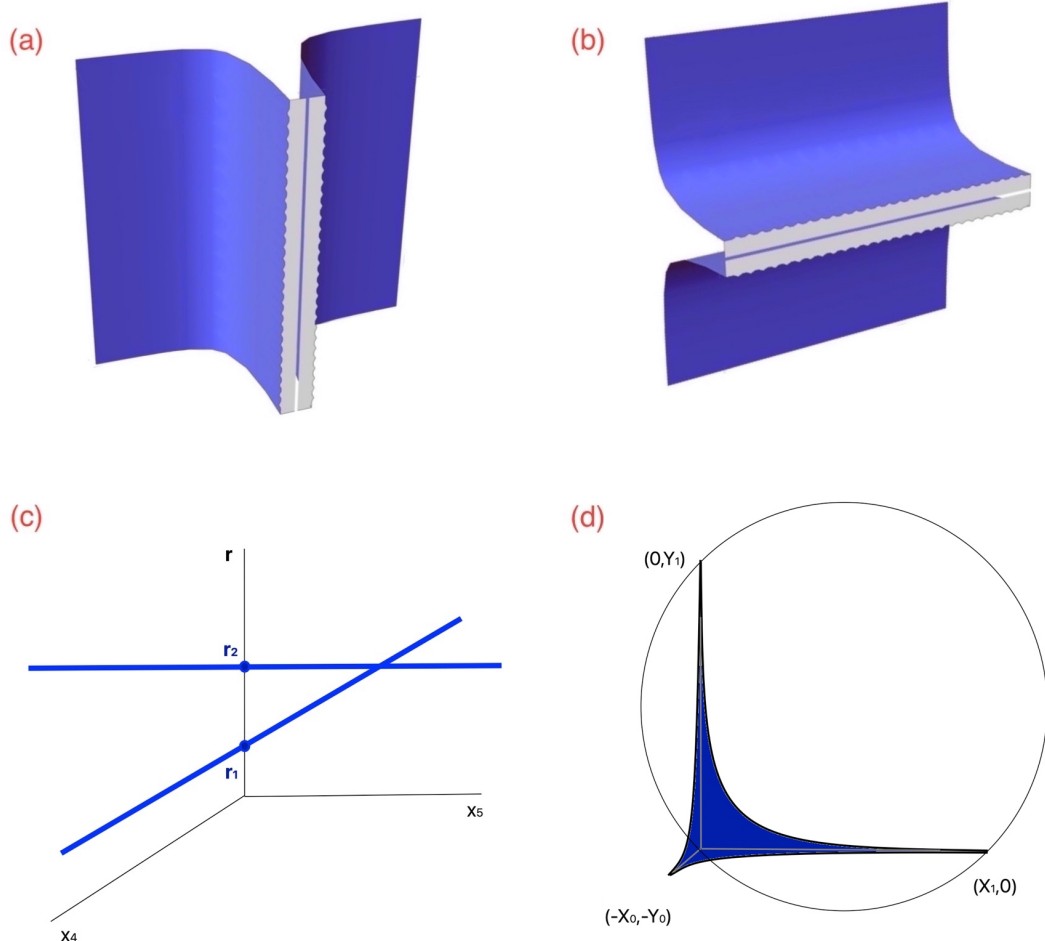

Figure 1: (a) and (b) depict the M5-brane ridge-shaped spike deformations $X \sim \frac{1}{\rho_X^2}$ and $Y \sim \frac{1}{\rho_Y^2}$ along the transverse $X$- and $Y$-directions respectively (the other worldvolume directions are suppressed). Fig.(c) shows the string soliton locations $\vec{\rho}_1 = (\vec{r}_1, x_1, 0)$ and $\vec{\rho}_2 = (\vec{r}_2, 0, x_2)$ in the M5-worldvolume (the strings are orthogonal, stretching along the $x^4$ and $x^5$ axes): the vertical axis represents the 3-dim subspace $\vec{r}$. The M5-brane prong is roughly a superposition of spikes (a), (b). Fig.(d) shows the shape in $(X, Y)$-transverse space of the M5 prong stretching out from the M5-brane at $(-X_0, -Y_0)$ (with individual spike-legs along $X$ and $Y$), in the limit $X_0, Y_0 \ll X_1, Y_1$, near the wall of marginal stability (circle).

## 3.2 String soliton bound states: M5-brane prongs

Now we describe field configurations given by two scalars on a single M5-brane: the M5-brane worldvolume deformation, roughly speaking, is now two-pronged. In an essential sense, this arises by trying to superpose two string solitons, each being the endline of an M2-brane ending on the M5-brane. These are stretched along two distinct non-parallel M5-worldvolume spatial directions, creating a superposition of two M5-brane spikes, one along the $X'$-direction and the other along the $Y'$-direction. Imposing boundary conditions on the scalar moduli in the vicinity of the string "cores" ensures that the resulting configuration is a molecule-like bound state of the two non-parallel string solitons with fixed transverse separation along the $\vec{r}$-direction, as we will see. Intuition for these boundary conditions arises from string web states in $\mathcal{N}{=}4$ SYM theories arising on D3-branes (a brief recap appears in Appendix B). We will call the resulting M5-brane deformations as M5-brane "prongs".

$(1,1)-(1,0)-(0,1)$ *prong*: The simplest such configuration is described by the $(1,1)-(1,0)-(0,1)$ state: in the D3-brane limit, this is described by the bound state of a fundamental $F1$- and $D1$-string ending on a D3-brane. This can be viewed as M-theory with the 45-directions compactified on a torus $T^2_{45}$: in the uncompactified M-theory description we have an M5-brane with two M2-branes stretched along say the $x_6$- and $x_7$-directions, and ending on lines stretching along the $x_4$- and $x_5$-directions in the M5-brane worldvolume: upon compactification the M5-brane becomes the D3-brane while the M2-branes stretched along $x_4$ and $x_5$ become respectively the F1- and D1-string.

The M5-M2 brane orientations are (suppressing directions that do not enter)

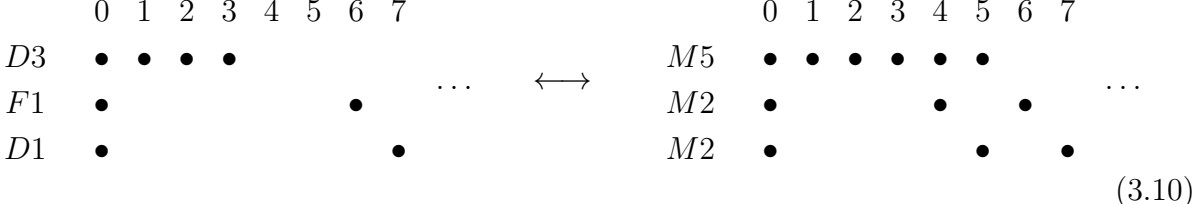

$$\tag{3.10}$$

In the D3-brane effective field theory, this is described as a bound state representing a $(1,1)$-charge molecule, with constituent charges $(1,0)$ and $(0,1)$: geometrically this can be described by a "prong web" obtained by deformations of the D3-brane worldvolume with two scalars. We expect and are looking for a similar prong-web description in the uncompactified M-theory, with two scalars characterizing nontrivial M5-brane worldvolume deformations representing a nontrivial bound state of the two string solitons resulting from the end-lines of the two M2-branes ending on the M5-brane.

We will take the strings to be stretched along the $x^4$ and $x^5$ directions but separated in the $\vec{r}$-directions (Figure 1 (c)): the $\zeta$-vectors and string coordinate parametrizations (2.18)

are

$$\zeta_1 = (1,0) : \qquad \vec{\rho}_1 = (\vec{r}, x^4, x^5)_1 = (\vec{r}_1, x_1, 0) \quad \forall\, x_1 \,;$$
$$\zeta_2 = (0,1) : \qquad \vec{\rho}_2 = (\vec{r}, x^4, x^5)_2 = (\vec{r}_2, 0, x_2) \quad \forall\, x_2 \,. \tag{3.11}$$

We have used the translation symmetry along the $x^5$ direction to set the locations $x_1^5$ and $x_2^4$ of the string solitons to be $x_1^5 = 0$ and $x_2^4 = 0$: in other words, we have taken the strings to lie along the $x^4$ and $x^5$ axes, without loss of generality. We expect this system can be described by the scalar field configurations (using the notation (2.1))

$$X = \frac{q_e}{|\vec{\rho}_X - \vec{\rho}_X^1|^2} - X_0 = \frac{q_e}{(\vec{r} - \vec{r}_1)^2 + (x^5)^2} - X_0 \,, \qquad\qquad \vec{\rho}_X \cdot \zeta_1 = 0 \,,$$

$$Y = \frac{q_m}{|\vec{\rho}_Y - \vec{\rho}_Y^2|^2} - X_0 = \frac{q_m}{(\vec{r} - \vec{r}_2)^2 + (x^4)^2} - Y_0 \,, \qquad\qquad \vec{\rho}_Y \cdot \zeta_2 = 0 \,,$$

$$\tilde{H}^{a4} \hat{x}^a = \vec{E} = \nabla X \,, \qquad \tilde{H}^{a5} \hat{x}^a = \vec{B} = \nabla Y \,, \qquad \tilde{H}^{45} = \Pi = \partial_4 Y - \partial_5 X \,. \tag{3.12}$$

These are roughly the superposition of an electric spike (3.5) along $X$ and a magnetic one along $Y$, as shown in Figure 1 (a) and (b). (Note that electric/magnetic here are simply labels for 4/5-directions of the strings which are self-dual solitons in 6-dimensions; upon 45-compactification, these will map to electric/magnetic point charges in 4-dimensions.) They can be seen to be a valid solution to the BPS bound equations (2.11) alongwith the line source conditions (2.15) with the $\zeta$-vectors above (and thereby also satisfy harmonicity (2.16) in the appropriate transverse 4-subspaces). We can rewrite the field configurations (3.12) as

$$X = \frac{q_e}{(\vec{r} - \vec{r}_1)^2 + (\zeta_1^\perp \cdot \vec{x})^2} - X_0 \,, \qquad Y = \frac{q_m}{(\vec{r} - \vec{r}_2)^2 + (\zeta_2^\perp \cdot \vec{x})^2} - Y_0 \,. \tag{3.13}$$

The distance elements here correspond to the distance in the subspace transverse to the string soliton, which thus is conveniently expressed using the unit normals $\zeta_1^\perp$, $\zeta_2^\perp$ to the string orientations $\zeta_1, \zeta_2$ (similar to (3.8)).

We will now examine the M5-brane deformation geometry described by these field configurations. Firstly, as we go out to infinity on the M5-brane, we expect the deformations (3.12) become vanishingly small, *i.e.*

$$\vec{\rho} \to \infty : \qquad (X, Y) \to (-X_0, -Y_0) \,, \tag{3.14}$$

which is the location of the M5-brane. The strings extend in one of the $x^4, x^5$-directions: if we go out to infinity in the 3-space directions, the deformations (3.12) scale as

$$\vec{r} \to \infty, \;\; x^4, x^5 \text{ finite} : \qquad X \sim \frac{q_e}{|\vec{r}|^2} - X_0 \,, \quad Y \sim \frac{q_m}{|\vec{r}|^2} - Y_0 \,. \tag{3.15}$$

Thus when

$$\frac{1}{q_e} X_0 = \frac{1}{q_m} Y_0 \ , \tag{3.16}$$

the M5-brane deformation in this asymptotic regime traces a line along $(1,1)$ in the $X,Y$-space transverse to the M5-brane in $q_e, q_m$-units. Likewise the electric and magnetic field components in (3.12) scale as $(\vec{E}, \vec{B}) \propto (1,1)$, thereby encoding charge $(1,1)$. This relation (3.16) on the asymptotic moduli also arises from an understanding of the effective tension of these configurations, as we will see in Sec. 3.3.

Let us now study the scalar field configurations as we approach string$_1$ stretched along $\vec{\rho}_1 = (\vec{r}_1, x_1, 0)$ in 3.11). We see that the $X$ scalar displays a spike as expected, similar to (3.5), (3.7). Looking at both scalars, we find in this limit,

$$X \sim \lim_{\vec{\rho} \to \vec{\rho}_1^+} \frac{q_e}{|\vec{\rho} - \vec{\rho}_1|^2} - X_0 \ \to \ X_1 \ ; \qquad Y \sim \frac{q_m}{(\vec{r}_1 - \vec{r}_2)^2 + (x_1)^2} - Y_0 \ . \tag{3.17}$$

$X_1$ is the cutoff value of the divergent $X$ spike, similar to (3.6). The $Y$ scalar on the other hand has no singularity anywhere. Since $x_1$ here labels the coordinate along string$_1$, if we go out to large $x_1$ along the string, the first term dies and we obtain $Y \to -Y_0$. However at $x_1 = 0$ we obtain

$$x_1 = 0 : \qquad Y \sim \frac{q_m}{(\vec{r}_1 - \vec{r}_2)^2} - Y_0 \ . \tag{3.18}$$

Likewise as we approach string$_2$ stretched along $\vec{\rho} = (\vec{r}_2, 0, x_2)$ in 3.11), we see that the $Y$ scalar displays a spike as expected, similar to (3.5), (3.7), but the $X$ scalar appears finite. The scalars approach

$$X \sim \frac{q_e}{(\vec{r}_1 - \vec{r}_2)^2 + (x_2)^2} - X_0 \ ; \qquad Y \sim \lim_{\vec{\rho} \to \vec{\rho}_2^+} \frac{q_m}{|\vec{\rho} - \vec{\rho}_2|^2} - Y_0 \ \to \ Y_1 \ . \tag{3.19}$$

$Y_1$ is the cutoff value of the $Y$ spike similar to (3.6), and $X$ exhibits no singularity anywhere. $x_2$ here labels the coordinate along string$_2$, so for large $x_2$ along the string, the first term dies and we obtain $X \to -X_0$. However at $x_2 = 0$ we obtain

$$x_2 = 0 : \qquad X \sim \frac{q_e}{(\vec{r}_1 - \vec{r}_2)^2} - X_0 \ . \tag{3.20}$$

These points (3.20), (3.18), are special: they lie on the closest point of approach between the two strings. The distance between a point on string$_1$ and any point on string$_2$ is given by $(\vec{r}_1 - \vec{r}_2)^2 + (x_1)^2 + (x_2)^2$ which is minimum when $x_1 = x_2 = 0$: projecting onto the $(x^4, x^5)$-plane, this is simply the intersection of the lines along $(x_1, 0)$ and $(0, x_2)$.

From similar bound state configurations arising from D3-brane prongs (Appendix B), we know that the separation between the constituent charge cores is fixed and scales as the inverse distance from the wall of marginal stability. Using this intuition, let us imagine that

the transverse distance $|\vec{r}_1 - \vec{r}_2|$ between the strings is fixed. It is then consistent to impose from (3.18), (3.20), that

$$\frac{1}{(\vec{r}_1 - \vec{r}_2)^2} = \frac{1}{q_e} X_0 = \frac{1}{q_m} Y_0 \ . \qquad (3.21)$$

Overall, this suggests the boundary conditions (2.19),

$$\vec{\rho} \to (\vec{r}_1, 0, 0): \quad (X, Y) \to (X_1, 0) \ ; \qquad \vec{\rho} \to (\vec{r}_2, 0, 0): \quad (X, Y) \to (0, Y_1) \ , \qquad (3.22)$$

imposed at the closest point of approach of the two orthogonal strings. These boundary conditions are equivalent to fixing the transverse separations between the strings as (3.21). As we have seen, the scalar moduli values vary in the various asymptotic M5-brane world-volume limits (3.15), (3.17), (3.18), (3.19), (3.20): these variations, summarized in Table 1, are consistent with (3.22) and follow from the field configurations (3.12).

| $(X, Y) \to (X_1, 0)$ | $\vec{r} \to \vec{r}_1$ | $x_1 = 0$ |
|---|---|---|
| $(X, Y) \to (X_1, -Y_0)$ | $\vec{r} \to \vec{r}_1$ | $|x_1| \to \infty$ |
| $(X, Y) \to (0, Y_1)$ | $\vec{r} \to \vec{r}_2$ | $x_2 = 0$ |
| $(X, Y) \to (-X_0, Y_1)$ | $\vec{r} \to \vec{r}_2$ | $|x_2| \to \infty$ |
| $(X, Y) \to (-X_0, -Y_0)$ | $\vec{r} \to \infty$ | $|\vec{x}| \to \infty$ |

Table 1: The $(X, Y)$ scalar moduli values at various regimes along the two strings.

Now, along with the transverse separations (3.21), we also obtain the inverse cross-sectional core sizes of the two string solitons from (3.17), (3.19),

$$\alpha_{ii} \equiv \lim_{\vec{\rho} \to \vec{\rho}_i^+} \frac{1}{|\vec{\rho} - \vec{\rho}_i|^2} \ ; \qquad \alpha_{11} = \frac{X_0 + X_1}{q_e} \ , \qquad \alpha_{22} = \frac{Y_0 + Y_1}{q_m} \ , \qquad (3.23)$$

similar to the cutoff core size in (3.7) for a single spike.

From (3.21) and (3.23), we see that

$$X_0, Y_0 \to 0: \qquad |\vec{r}_1 - \vec{r}_2| \to \infty \ , \qquad \alpha_{11}^{-1} \sim \frac{q_e}{X_1} \ , \qquad \alpha_{22}^{-1} \sim \frac{q_m}{Y_1} \ , \qquad (3.24)$$

*i.e.* the constituent string solitons are infinitely separated in the 3-space directions transverse to their extended directions, while their core sizes $\alpha_{ii}^{-1}$ are finite (and small if $X_1, Y_1$ are large). Thus the bound state becomes arbitrarily loosely bound, *i.e.* it decays and disappears from the spectrum. There is no physical bound state solution with $(\vec{r}_1 - \vec{r}_2)^2 > 0$ to (3.21) for $X_0 < 0$ (the other side of the wall).

From Table 1 summarizing the moduli variations, we note from (3.19) that the M5-brane $X$-deformation along string$_2$ varies from $X = 0$ at $x_2 = 0$ to $X = -X_0$ at large $x_2$, and

likewise the M5-brane $Y$-deformation along string$_1$ interpolates from $Y = 0$ at $x_1 = 0$ to $Y = -Y_0$ at large $x_1$. Thus the M5-brane bending is nontrivial as we go out to infinity along the strings. Note that this does not happen for a single string soliton obtained from a single M2-brane ending on the M5-brane (there are some qualitative parallels with the M5-brane descriptions of $\mathcal{N}=2$ theories [54], although the brane bending is less severe in the present discussions). Overall however, in the limit of the M5-brane approaching the wall of marginal stability

$$X_0, Y_0 \ll X_1, \; Y_1 \; , \tag{3.25}$$

this variation of $X, Y$ is negligible compared with the individual spike legs which are much larger. Thus in this limit $X_0, Y_0 \ll X_1, Y_1$, the shape of the M5-brane in the transverse $X, Y$-space resembles Figure 1 (d): this refinement of the scalar boundary conditions and their asymptotics is similar to that in [19] for D3-brane SYM theories (but with additional intricacies due to the string-nature of the constituents). There are also parallels in the qualitative picture of the decay across the wall of marginal stability: wall-crossing amounts to the state becoming arbitrarily loosely bound, and is qualitatively similar to wall-crossing for black holes [36]. The limit $X_0, Y_0 \to 0$ of approaching the wall of marginal stability (circle in Fig. 1 (d)) means the shortest leg approaches zero size, *i.e.* the M5-brane at $(-X_0, -Y_0)$ approaches $(0,0)$.

It is also instructive to note that far from both of the string solitons, we obtain

$$X \sim \frac{q_e}{(\vec{r})^2 + (x_2)^2} - X_0 \; , \qquad Y \sim \frac{q_m}{(\vec{r})^2 + (x_1)^2} - Y_0 \; , \tag{3.26}$$

and the field configurations appear to describe two string solitons intersecting at $(x_1, x_2) = (0, 0)$. In other words the transverse $\vec{r}$-separation is not discernable and the two M2-branes now appear to approximately intersect each other on the M5-brane when viewed from afar. Note that this appears slightly different from the D3-brane limit where the two pointlike cores appear to effectively coalesce: the string solitons being extended objects along orthogonal directions remain orthogonal and extended.

Another interesting point, using the relation (2.4) between the worldvolume scalars in the field theory regime and the M5-brane coordinates, is

$$(\vec{r}_1 - \vec{r}_2)^2 \sim \frac{1}{X_0} \sim \frac{l_{11}^3}{x_0} \; . \tag{3.27}$$

If we consider holding the M5-brane coordinate distance $x_0$ fixed, taking $l_{11} \to 0$ implies that $|\vec{r}_1 - \vec{r}_2| \to 0$ generically, which suggests a junction-like configuration: the exception occurs when $x_0 = 0$ beyond which point the bound state ceases to exist. This appears to pertain to comments on M2-brane junctions in *e.g.* [38] and [39], as well as [16]. On the other hand, we have been describing the field theory regime, which arises by taking $l_{11} \to 0$

holding the "field theory distance" $X_0$ fixed (essentially, as in [9]). In this limit we obtain a smooth description of the string soliton molecule decaying at the wall of marginal stability as $X_0 \to 0$ (with no "junction" per se). Something similar occurs for D3-branes and string webs in the field theory regime, as discussed in [19, 20].

### 3.2.1   M5-brane prong for $(n, m) - (n, 0) - (0, m)$

Along very similar lines, we can describe the $(n, m) - (n, 0) - (0, m)$ M5-brane prong web representing a charge-$(n, m)$ string soliton molecule, with constituent string solitons of electric and magnetic charges $(n, 0)$ and $(0, m)$. The $\zeta$-vectors are again orthogonal, as in (3.11) with $\zeta_1 = (1, 0)$, $\zeta_2 = (0, 1)$, so the scalar field configurations again satisfy $X' = X$, $Y' = Y$. This is a 2-parameter family of string soliton bound states but the description is very similar overall to the case we just described, with $n = 1$, $m = 1$, so we will be brief here. The scalar field configurations are

$$X = \frac{n\, q_e}{(\vec{r} - \vec{r_1})^2 + (x^5)^2} - X_0 \;, \qquad Y = \frac{m\, q_m}{(\vec{r} - \vec{r_2})^2 + (x^4)^2} - Y_0 \;, \tag{3.28}$$

very similar to (3.12). The $(X, Y)$-moduli asymptotically, as $\vec{\rho} \to \infty$, trace out a line along $(n, m)$ in the moduli space when

$$\frac{m}{q_e} X_0 = \frac{n}{q_m} Y_0 \;. \tag{3.29}$$

Imposing the boundary conditions (3.22) on the scalars leads to

$$\frac{1}{(\vec{r_1} - \vec{r_2})^2} = \frac{1}{nq_e} X_0 = \frac{1}{mq_m} Y_0 \;, \tag{3.30}$$

as the transverse separation between the two orthogonal string solitons in the M5-brane worldvolume. We then also obtain the inverse core sizes

$$\alpha_{11} = \frac{X_0 + X_1}{nq_e} \;, \qquad \alpha_{22} = \frac{Y_0 + Y_1}{mq_m} \;. \tag{3.31}$$

The asymptotic behaviour of the field configurations, as well as the picture of the decay as the M5-brane approaches the wall of marginal stability, is very similar to the $(1, 1) - (1, 0) - (0, 1)$ case discussed earlier.

## 3.3   1-string tension, marginal stability and wall-crossing

We will now calculate the tension of the string soliton bound states we have been describing. It is useful to note that the $\zeta$-vectors we have been using admit a rotation by an overall angle $\varphi$ (which is just an overall rotation of the $x^4$-$x^5$ axes): this gives a slightly more general representation of the orthogonal $\zeta$-vectors in (3.11) as

$$\zeta^1 = \cos\varphi\, \hat{x}^4 - \sin\varphi\, \hat{x}^5 \;, \qquad \zeta^2 = \sin\varphi\, \hat{x}^4 + \cos\varphi\, \hat{x}^5 \;. \tag{3.32}$$

while retaining the geometry and physics of the field configurations. Then using (2.11), the Bogomolny completion (2.12) becomes (with $\zeta^1 \cdot \zeta^2 = 0$)

$$
\begin{aligned}
\mathcal{E} = {} & \tfrac{1}{2}\big|\vec{E} - \cos\varphi\nabla X - \sin\varphi\nabla Y\big|^2 + \tfrac{1}{2}\big|\vec{B} + \sin\varphi\nabla X - \cos\varphi\nabla Y\big|^2 \\
& + \tfrac{1}{2}\big|\Pi + \cos\varphi(\partial_5 X - \partial_4 Y) + \sin\varphi(\partial_4 X + \partial_5 Y)\big|^2 \\
& + \tfrac{1}{2}\big(\sin\varphi(\partial_5 X - \partial_4 Y) - \cos\varphi(\partial_4 X + \partial_5 Y)\big)^2 \\
& + \vec{E}\cdot(\cos\varphi\nabla X + \sin\varphi\nabla Y) + \vec{B}\cdot(-\sin\varphi\nabla X + \cos\varphi\nabla Y) \\
& - \Pi\big(\partial_5(\cos\varphi X + \sin\varphi Y) - \partial_4(-\sin\varphi X + \cos\varphi Y)\big) \\
& - (\partial_4 X \partial_5 Y - \partial_5 X \partial_4 Y)
\end{aligned}
\tag{3.33}
$$

We will now show that the cross-terms $\partial_4 X \partial_5 Y - \partial_5 X \partial_4 Y$ in (3.33) upon integrating over 5-space have negligible contribution on-shell. With nonzero $\varphi$, the expressions for the $X$ and $Y$ fields using (3.13) with (3.32) become

$$
X = \frac{q_e}{(\vec{r} - \vec{r}_1)^2 + (\sin\varphi\, x^4 + \cos\varphi\, x^5)^2} - X_0\,, \qquad Y = \frac{q_m}{(\vec{r} - \vec{r}_2)^2 + (\cos\varphi\, x^4 - \sin\varphi\, x^5)^2} - Y_0\,.
\tag{3.34}
$$

The cross term under the 5-space integral is

$$
\int d^3 r\, dx^4 dx^5\, (\partial_4 X \partial_5 Y - \partial_5 X \partial_4 Y)
\tag{3.35}
$$

It is convenient to consider an overall rotation by $\varphi$ in the $x^4 - x^5$ plane by defining

$$
x'^4 = \cos\varphi\, x^4 - \sin\varphi\, x^5\,, \qquad x'^5 = \sin\varphi\, x^4 + \cos\varphi\, x^5\,.
\tag{3.36}
$$

Then (3.34) takes a simpler form

$$
X = \frac{q_e}{(\vec{r} - \vec{r}_1)^2 + (x'^5)^2} - X_0\,, \qquad Y = \frac{q_m}{(\vec{r} - \vec{r}_2)^2 + (x'^4)^2} - Y_0\,,
\tag{3.37}
$$

and the form of the cross-terms in (3.35) remains the same[1]

$$
\int d^3 r\, dx'^4 dx'^5\, (\partial_{4'} X \partial_{5'} Y - \partial_{5'} X \partial_{4'} Y)
\tag{3.38}
$$

The first term $\partial_{4'} X \partial_{5'} Y$ vanishes and the second term upon integration by parts gives

$$
-\int d^3 r\, dx'^4 \left[ X \partial_{4'} Y\Big|_{x'^5=0^+}^{\infty} + X \partial_{4'} Y\Big|_{-\infty}^{x'^5=0^-} \right]
\tag{3.39}
$$

---

[1]Essentially these terms have the structure of the corresponding terms in the wedge-product $\int dX \wedge dY$ so they are manifestly unchanged under the coordinate transformation. This can also be checked using $\partial_4 = \cos\varphi\frac{\partial}{\partial x'^4} + \sin\varphi\frac{\partial}{\partial x'^5}$ and $\partial_5 = -\sin\varphi\frac{\partial}{\partial x'^4} + \cos\varphi\frac{\partial}{\partial x'^5}$ and evaluating explicitly.

In the above, we have introduced boundary values near $x^5 = 0$ value due to string$_1$ when $\vec{r} = \vec{r}_1$. When $\vec{r} \neq \vec{r}_1$ the effect of doing this in the integral expression is negligible.

$\partial_{4'} Y$ can be factored out here since it is independent of $x'^5$ and another integration by parts can be done here $w.r.t.$ $dx'^4$ giving

$$- \int d^3 r \left[ X \Big|_{x'^5=0^+}^{\infty} + X \Big|_{-\infty}^{x'^5=0^-} \right] \left[ Y \Big|_{x'^4=0^+}^{\infty} + Y \Big|_{-\infty}^{x'^4=0^-} \right] \tag{3.40}$$

Since the $X$ and $Y$ fields do not depend on the signs of $x'^5$ and $x'^4$ coordinates, we conclude that the square bracket terms vanish. Note that this vanishing of these cross-terms (last line in (3.33)) is on-shell, using (3.37).

Now, the terms in lines 4 and 5 in (3.33) upon integrating over 5-space can be recast as

$$\int dx^4 \int d^3 r dx^5 \, \tilde{\mathcal{H}}^{a4} \left( \cos \varphi \, \partial_a X + \sin \varphi \, \partial_a Y \right) + \int dx^5 \int d^3 r dx^4 \, \tilde{\mathcal{H}}^{a5} \left( - \sin \varphi \, \partial_a X + \cos \varphi \, \partial_a Y \right) \tag{3.41}$$

which upon doing the integration by parts and subsequently using the equation of motion $\partial_a \tilde{\mathcal{H}}^{ab} = 0$ become

$$\int dx^4 \oint_{S^3} d\hat{s}^a \, \tilde{\mathcal{H}}^{a4} \left( \cos \varphi \, X + \sin \varphi \, Y \right) + \int dx^5 \oint_{S^3} d\hat{s}^a \, \tilde{\mathcal{H}}^{a5} \left( - \sin \varphi \, X + \cos \varphi \, Y \right). \tag{3.42}$$

We are considering two strings with charges

$$(Q_E^i, Q_B^i), \ i = 1, 2 : \qquad Q_E^i = \oint_{S_{X,i}^3} \tilde{H}^{a4} ds^a \ , \qquad Q_B^i = \oint_{S_{Y,i}^3} \tilde{H}^{a5} ds^a \ , \tag{3.43}$$

with scalar moduli values at their cores $(X_i', Y_i')$. At spatial infinity far from both strings, charge conservation gives $(Q_E^0, Q_B^0) = - \sum_i (Q_E^i, Q_B^i)$ and the moduli values are $(-X_0, -Y_0)$. For a single string, the asymptotic boundary to the 4-dim transverse space is $S^3$. For the combined configuration of two strings we are discussing, integrating over 5-space requires incorporating the appropriate $S_{X,i}^3$ and $S_{Y,i}^3$, giving

$$L \sum_i \cos \varphi \big( (X_i + X_0) Q_E^i + (Y_i + Y_0) Q_B^i \big) + \sin \varphi \big( (Y_i + Y_0) Q_E^i - (X_i + X_0) Q_B^i \big) \ , \tag{3.44}$$

with $L \equiv \int dx^4 \sim \int dx^5$ the regulated length of both strings. Extremizing w.r.t. $\varphi$ gives

$$T \equiv \frac{M}{L} = \sqrt{\big( (X_i + X_0) Q_E^i + (Y_i + Y_0) Q_B^i \big)^2 + \big( (Y_i + Y_0) Q_E^i - (X_i + X_0) Q_B^i \big)^2} \tag{3.45}$$

as the string tension and

$$\tan \varphi = \frac{(Y_i + Y_0) Q_E^i - (X_i + X_0) Q_B^i}{(X_i + X_0) Q_E^i + (Y_i + Y_0) Q_B^i} \tag{3.46}$$

In the simplest case of one electric string and one magnetic string, we have $(Q_E^1, Q_B^1) = (q_e, 0)$ and $(Q_E^2, Q_B^2) = (0, q_m)$, and the moduli values are $(X_1, 0)$ and $(0, Y_1)$, so

$$\tan\varphi = \frac{(0 + Y_0)Q_E^1 - (0 + X_0)Q_B^2}{(X_1 + X_0)Q_E^1 + (Y_1 + Y_0)Q_B^2} \to 0 \quad \Rightarrow \quad \frac{1}{q_e}X_0 = \frac{1}{q_m}Y_0 \ , \tag{3.47}$$

which is the moduli constraint (3.16) encoding the asymptotic shape of the M5-brane deformation. This then implies

$$T = (X_1 + X_0)q_e + (Y_1 + Y_0)q_m \ . \tag{3.48}$$

This effective tension can be interpreted in a simple way from the geometry in Figure 1 (d), as the sum of the tensions of a 3-pronged M2-brane web comprising

$$
\begin{array}{llll}
\text{leg}_0\,, & \text{charge}(1,1): & (-X_0, -Y_0) \to (0,0) \ , \\
\text{leg}_1\,, & \text{charge}(1,0): & (0,0) \to (X_1, 0) \ , \\
\text{leg}_2\,, & \text{charge}(0,1): & (0,0) \to (0, Y_1) \ , & \tag{3.49}
\end{array}
$$

Taking the membrane tension of a $(m,n)$ charge M2-brane as $\sqrt{m^2 q_e^2 + n^2 q_m^2}$, the tension of the composite M2-brane web above, comprising legs$_{0,1,2}$, is

$$T = \sqrt{q_e^2 + q_m^2}\sqrt{X_0^2 + Y_0^2} + \sqrt{q_e^2 + 0}\,X_1 + \sqrt{0 + q_m^2}\,Y_1 \tag{3.50}$$

which is identical to (3.48) using (3.16). We note that as a 1-string state tension, $T \leq T_1 + T_2$, *i.e.* $T$ is always less than the 2-string tension

$$T_1 + T_2 = q_e\sqrt{(X_1 + X_0)^2 + (Y_0)^2} + q_m\sqrt{(X_0)^2 + (Y_1 + Y_0)^2} \ , \tag{3.51}$$

which is obtained from the lengths of two independent M2-branes stretched between the $(X, Y)$-locations $\{(-X_0, -Y_0),\ (X_1, 0)\}$ and $\{(-X_0, -Y_0),\ (0, Y_1)\}$ respectively (see the transverse geometry in Figure 1 (d)). The location in moduli space where the 1-string state becomes equal in tension to the 2-string state is the wall of marginal stability (represented by the circle in $X, Y$-space in Figure 1 (d)):

$$X_0, Y_0 \to 0 : \quad T \to T_1 + T_2 \ . \tag{3.52}$$

This confirms the decay process of the string soliton bound state into the 2-string state at the wall of marginal stability that we discussed around (3.24) in terms of the field configurations earlier.

## 3.4   More on Bogomolny and the 1-string tension on-shell

In the previous section 3.3, we have seen that the effective 1-string tension obtained upon integrating the energy functional (2.5) acquires a relatively simple form near marginal stability, as in eq. (3.50). In particular, note that the energy functional (2.5) is not written in strict Bogomolny form (perfect squares plus boundary terms off-shell, such as in eq.(B.1) in the D3-brane case reviewed in App. B) since there are extra cross-terms, which however give vanishing contribution on-shell in the vicinity of marginal stability. We will discuss this in a slightly more general way now, noting that near marginal stability we have two well-separated constituent strings that are loosely bound.

The energy functional (2.5) contained two sets of cross-terms: the second set containing $\zeta^1 \cdot \zeta^2$ vanishes when the constituent strings are orthogonal. The first set of cross-terms can be reorganized as

$$\sum_{a \neq b} \zeta_a^1 \zeta_b^2 \left( \partial_a X \partial_b Y - \partial_b X \partial_a Y \right) = \sum_{a \neq b} \left( \zeta_a^1 \zeta_b^2 - \zeta_b^1 \zeta_a^2 \right) \partial_a X \partial_b Y$$

$$= \sum_{a \neq b} \left[ \partial_b \left( \left( \zeta_a^1 \zeta_b^2 - \zeta_b^1 \zeta_a^2 \right) (\partial_a X) Y \right) - \left( \zeta_a^1 \zeta_b^2 - \zeta_b^1 \zeta_a^2 \right) (\partial_b \partial_a X) Y \right]. \qquad (3.53)$$

The second set of terms here vanishes (summing symmetric and antisymmetric pieces), leaving a total derivative term. Integrating the first set of terms over 5-space with boundaries being $S^3_\infty$ and the two $S^3$s enclosing each of the two (well-separated) strings gives

$$\sum_{a \neq b} \int d^5 x \, \partial_b \left( \left( \zeta_a^1 \zeta_b^2 - \zeta_b^1 \zeta_a^2 \right) (\partial_a X) Y \right)$$

$$= \int dl_{\text{string}_1} \sum_{a \neq b} \int \left( d\hat{S}^3 \right)^b \left( \left( \zeta_a^1 \zeta_b^2 - \zeta_b^1 \zeta_a^2 \right) (\partial_a X) Y \right) \Big|_{\text{string}_1}$$

$$+ \int dl_{\text{string}_2} \sum_{a \neq b} \int \left( d\hat{S}^3 \right)^b \left( \left( \zeta_a^1 \zeta_b^2 - \zeta_b^1 \zeta_a^2 \right) (\partial_a X) Y \right) \Big|_{\text{string}_2} \qquad (3.54)$$

since the $S^3$ at infinity is at constant modulus value $X = -X_0$ (so its derivative vanishes). Each string in 5-dimensions is akin to a point particle in the transverse 4-dimensions: we draw a 3-sphere $S^3$ surrounding the string. The measure $dl_{\text{string}_i}$ is the length element along the string $i$ with $i = 1, 2$, and $(d\hat{S}^3)^b$ are the differential 3-volume elements transverse to the strings, and the $b$-index indicates its orientation in the 4-dim spatial boundary defined surrounding the respective strings. The string orientations are defined by the $\zeta$-vectors which lie in the $(x^4, x^5)$-plane and the $b$-index takes values accordingly.

Near marginal stability we have $X_0, Y_0$ small: the strings are well-separated. We now

recast the above boundary terms as

$$\sum_{a \neq b} \int d^5x \, \partial_b \Big( \big( \zeta_a^1 \zeta_b^2 - \zeta_b^1 \zeta_a^2 \big) (\partial_a X) Y \Big)$$

$$= \int dl_{\text{string}_1} \sum_{a \neq b_{\perp_1}} \int \big( d\hat{S}^3 \big)^{b=b_{\perp_1}} \Big( \big( \zeta_a^1 \zeta_{b_{\perp_1}}^2 - \zeta_{b_{\perp_1}}^1 \zeta_a^2 \big) (\partial_a X) Y \Big) \Big|_{\text{string}_1}$$

$$+ \int dl_{\text{string}_2} \sum_{a \neq b_{\perp_2}} \int \big( d\hat{S}^3 \big)^{b=b_{\perp_2}} \Big( \big( \zeta_a^1 \zeta_{b_{\perp_2}}^2 - \zeta_{b_{\perp_2}}^1 \zeta_a^2 \big) (\partial_a X) Y \Big) \Big|_{\text{string}_2} \quad (3.55)$$

Here $b_\perp$ indicates the direction in the $(x^4, x^5)$-plane orthogonal to the respective strings.

Analysing further, in the string-1 terms (second line above), the component $\zeta_{b_{\perp_1}}^1$ is zero (since $\zeta^1$ has no component orthogonal to string$_1$), and further $\zeta_a^1 \partial_a X = 0$ from translation invariance along string-1. Thus these string-1 terms in the second line vanish.

Likewise in the string-2 terms (third line above), the component $\zeta_{b_{\perp_2}}^2$ is zero (since $\zeta^2$ has no component orthogonal to string$_2$). The remaining term can be rewritten as

$$- \sum_{a \neq b_{\perp_2}} \int \big( d\hat{S}^3 \big)^{b=b_{\perp_2}} \int dl_{\text{string}_2} \Big( \big( \zeta_{b_{\perp_2}}^1 \zeta_a^2 \big) (\partial_a X) \Big) \Big|_{\text{string}_2} \times Y_1 \,. \quad (3.56)$$

(recall that $Y_1$ is a constant cut-off value for the scalar $Y$ near the location of string$_2$) Now we note that $X$ is an even function along string-2 (symmetric about the point of closest approach of string-2 with string-1). Thus its derivative $\zeta_a^2 \partial_a X$ is an odd function along string-2: so integrating along the entire length of string-2 gives a vanishing contribution. Thus this term also vanishes.

Let us consider the simplest case of two orthogonal strings to illustrate how the above arguments work. To compare with our analysis in sec. 3.3, recall that (from (3.12)) $\partial_4 X = 0$, and $X = \frac{q}{|\vec{r} - \vec{r}_1|^2 + (x^5)^2} - X_0$. Thus the first set of terms gives

$$\int dl_{\text{string}_1} \int \big( d\hat{S}^3 \big)^{b_{\perp_1}=5} \Big( \big( \zeta_4^1 \zeta_5^2 \big) (\partial_4 X) Y \Big) \Big|_{\text{string}_1} \quad (3.57)$$

which thus vanishes. Note that the $S^3$ here surrounds string$_1$ embedded in the $\mathbb{R}^4$ orthogonal to the $x^4$-direction, so that only $b_{\perp_1} = 5$ appears in the $S^3$-element above. Likewise, the terms (3.56) here become

$$- \int \big( d\hat{S}^3 \big)^{b_{\perp_2}=4} \int dx^5 \Big( \big( \zeta_{b_{\perp_2}=4}^1 \zeta_5^2 \big) (\partial_5 X) \Big) \Big|_{\text{string}_2} \times Y_1 \,, \quad (3.58)$$

and $\partial_5 X$ is odd, so the integral vanishes.

Thus overall, we find these extra cross-terms in the Bogomolny completion vanish on-shell, when integrated over the 5-space with appropriate boundaries around the two well-separated strings near marginal stability (when the constituent strings are not orthogonal

there are further terms as well, the $\zeta^1 \cdot \zeta^2$ terms in (2.5)). Of course, in effect we have simply recast the earlier analysis leading to (3.50) in a slightly more general manner, but this is perhaps useful in illustrating these extra terms as vanishing boundary terms on-shell. The essential ingredients here of course is that the strings are well-separated so one can break up the 5-space integral into boundary terms over appropriate separate $S^3$s surrounding the two strings. This is of course justified near marginal stability and becomes increasingly accurate here, and is consistent with the fact that our entire analysis is only reliable near marginal stability. However it begs the question of finding "strict Bogomolny completions" that are manifestly of the form "perfect-squares plus boundary terms" but at an *off-shell* level.

From the point of view of regarding one string as background and the other as a probe, we expect that the probe string will feel larger 3-form and scalar forces near its "bulk" which is closer to the bulk of the background string. This suggests that it will bend resulting in curved string profiles, unlike the straight string configurations (Figure 1 (c)) we have been studying. These bent (or curved) string equilibrium configurations will likely be solutions to different BPS equations following from some possibly different Bogomolny energy extremization, holding different boundary terms fixed, with no extra cross-terms (looking for such bent string configurations appears difficult however!). Near marginal stability, the strings are well-separated and so will straighten out (with negligible bending), thus coinciding with our picture of straight non-parallel loosely bound strings near marginal stability. In this light, it is perhaps not surprising that these cross-terms above only vanish on-shell.

# 4   More general M5-M2 prong states

We now consider more general M5-M2 prong bound states: in general these have charges $(p_1+p_2, q_1+q_2) - (p_1, q_1) - (p_2, q_2)$ and corresponding nontrivial angles between the constituent legs. The field configurations can be constructed using two general $\zeta$-vectors with $X', Y'$ defined appropriately as in (2.11). We will however find it more instructive to discuss these illustrating them with two concrete nontrivial examples.

## 4.1   $(2, 1) - (1, 0) - (1, 1)$

Consider the charges $(2, 1) - (1, 0) - (1, 1)$. We will keep the factors $q_e, q_m$ explicit. This configuration is captured by the $\zeta$-vectors and the string parametrizations via $(x^4, x^5)$-coordinates,

$$\zeta_1 = (1, 0): \qquad \vec{x}_1 \equiv (x^4, x^5)_1 = x_1(1, 0) \quad \forall\, x_1, \quad \vec{r} = \vec{r}_1, \qquad (4.1\text{a})$$

$$\zeta_2 = \frac{1}{\sqrt{1 + \frac{q_e^2}{q_m^2}}} \left( \frac{q_e}{q_m}, 1 \right) \equiv \gamma(\beta, 1): \qquad \vec{x}_2 \equiv (x^4, x^5)_2 = x_2(\beta, 1) \quad \forall\, x_2, \quad \vec{r} = \vec{r}_2. \qquad (4.1\text{b})$$

We have used the translation symmetry along the $x^4$ and $x^5$ directions to parametrize the string coordinates so that they always pass through the origin $(x^4, x^5) = (0, 0)$. Projecting these lines onto the $(x^4, x^5)$-plane, these lines can always be seen to intersect at the origin which is their closest point of approach in this convenient parametrization.

Using the "basis" field configurations (2.17) with $c_Y \equiv q_m/\gamma$ and $c_X \equiv q_e$, the BPS equations (2.11) give the solutions

$$X' = \frac{q_e}{(\vec{r} - \vec{r}_1)^2 + (\zeta_1^\perp \cdot (\vec{x} - \vec{x}_1))^2} + \frac{q_e}{(\vec{r} - \vec{r}_2)^2 + (\zeta_2^\perp \cdot (\vec{x} - \vec{x}_2))^2} - X_0' ,$$
$$Y' = \frac{q_m}{(\vec{r} - \vec{r}_2)^2 + (\zeta_2^\perp \cdot (\vec{x} - \vec{x}_2))^2} - Y_0' . \tag{4.2}$$

The denominators here encode the transverse distance from any point on string$_1$ or string$_2$: since we have translation invariance along the strings, this means the "equipotential surfaces" for string$_1$ are concentric cylinders around it, and likewise for string$_2$. Thus if we translate along the string, the corresponding field configuration must remain constant so the location along the string should scale away from these expressions: this can be seen explicitly by noting that $\zeta_1^\perp \cdot \vec{x}_1 = 0$ and $\zeta_2^\perp \cdot \vec{x}_2 = 0$ using the parametrizations (4.1). This gives

$$X' = \frac{q_e}{(\vec{r} - \vec{r}_1)^2 + (\zeta_1^\perp \cdot \vec{x})^2} + \frac{q_e}{(\vec{r} - \vec{r}_2)^2 + (\zeta_2^\perp \cdot \vec{x})^2} - X_0' ,$$
$$Y' = \frac{q_m}{(\vec{r} - \vec{r}_2)^2 + (\zeta_2^\perp \cdot \vec{x})^2} - Y_0' . \tag{4.3}$$

These then give the electric and magnetic components $\vec{E}, \vec{B}$ of the field strength. As $\vec{r} \to \infty$ we have

$$\vec{r} \to \infty, \quad x^4, x^5 \text{ finite}: \qquad X' \sim \frac{2q_e}{|\vec{r}|^2} - X_0' , \qquad Y' \sim \frac{q_m}{|\vec{r}|^2} - Y_0' , \tag{4.4}$$

so we find $(\vec{E}, \vec{B}) = (\nabla X', \nabla Y') \sim (2, 1)$ encoding the asymptotic charge of this configuration. Thus this M5-brane deformation in this asymptotic regime traces a line along $(2, 1)$ in the $(X', Y')$ moduli space transverse to the M5-brane when

$$\frac{1}{q_e} X_0' = \frac{2}{q_m} Y_0' . \tag{4.5}$$

Now we study the scalar field configurations as we approach the strings. First, analogous to (3.22), let us impose the boundary conditions (2.19). Then from the $Y'$ configuration, we have $Y' \to 0$ which gives

$$\frac{1}{(\vec{r}_1 - \vec{r}_2)^2} = \frac{1}{q_m} Y_0' = \frac{1}{2q_e} X_0' . \tag{4.6}$$

This holds at $(x^4, x^5) = (0, 0)$. Away from this, as we go along string$_1$ stretched along $\vec{\rho} = (\vec{r}_1, x_1, 0)$, the $Y'$ scalar behaves as

$$Y' \sim \frac{q_m}{(\vec{r}_1 - \vec{r}_2)^2 + \gamma^2 x_1^2} - Y_0' \xrightarrow{|x_1| \to \infty} -Y_0' , \tag{4.7}$$

using $\zeta_2^\perp = \gamma(1, -\beta)$. Likewise one can analyse the $X', Y'$ field configurations in various asymptotic limits, as we go along string$_1$ or string$_2$. This is most efficiently done using the relations (2.11) in terms of the "basis fields" $X, Y$: using (4.1) these become

$$X' = X + \beta\gamma Y , \qquad Y' = \gamma Y , \tag{4.8}$$

so

$$Y_0' = \gamma Y_0 , \quad X_0' = X_0 + \beta\gamma Y_0 , \quad X_1' = X_1 , \quad Y_1' = \gamma Y_1 . \tag{4.9}$$

The resulting scalar asymptotics on the moduli space for $X', Y'$, in various limits can be written by using those for $X, Y$ in Appendix C and are tabulated in Table 2.

| $(X', Y') \to (X_1', 0)$ | $\vec{r} \to \vec{r}_1$ | $x_1 = 0$ |
|---|---|---|
| $(X', Y') \to (X_1' - \beta Y_0' , -Y_0')$ | $\vec{r} \to \vec{r}_1$ | $|x_1| \to \infty$ |
| $(X', Y') \to (\beta Y_1' , Y_1')$ | $\vec{r} \to \vec{r}_2$ | $x_2 = 0$ |
| $(X', Y') \to (-(X_0' - \beta Y_0') + \beta Y_1' , Y_1')$ | $\vec{r} \to \vec{r}_2$ | $|x_2| \to \infty$ |
| $(X', Y') \to (-X_0', -Y_0')$ | $\vec{r} \to \infty$ | $|x_1|, |x_2| \to \infty$ |

Table 2: $X', Y'$ moduli asymptotics

It is clear that in the limit of approaching the wall of marginal stability

$$X_0', \ Y_0' \ \ll \ X_1', \ Y_1' , \tag{4.10}$$

the M5-brane bendings "straighten out" and the moduli approach (i) $(X_1', 0)$ along the entire string$_1$, $i.e.$ as $\vec{\rho} \to (\vec{r}_1, x_1, 0)$, and (ii) $(\beta Y_1', Y_1')$ along the entire string$_2$, $i.e.$ as $\vec{\rho} \to (\vec{r}_2, \beta x_2, x_2)$. This is similar to the results in the previous section where we discussed the $(1, 1) - (1, 0) - (0, 1)$ M5-brane prongs.

## 4.2  $(3, 2) - (1, 1) - (2, 1)$

The analysis for this example is similar to the previous case but is a bit more involved. This configuration has the $\zeta$-vectors and the coordinate parametrizations on the strings,

$$\zeta_1 = \frac{1}{\sqrt{1 + \frac{q_m^2}{q_e^2}}} \left(1 , \frac{q_m}{q_e}\right) \equiv \gamma_1(1 , \beta_1) : \qquad \vec{x}_1 \equiv (x^4, x^5)_1 = x_1(1 , \beta_1) \quad \forall \ x_1 , \quad \vec{r} = \vec{r}_1 , \tag{4.11a}$$

$$\zeta_2 = \frac{1}{\sqrt{1 + \frac{(2q_e)^2}{q_m^2}}} \left(\frac{2q_e}{q_m} , 1\right) \equiv \gamma_2(\beta_2 , 1) : \qquad \vec{x}_2 \equiv (x^4, x^5)_2 = x_2(\beta_2, 1) \quad \forall \ x_2 , \quad \vec{r} = \vec{r}_2 . \tag{4.11b}$$

Using the field configurations in (2.17) with $c_X \equiv q_e/\gamma_1$ and $c_Y \equiv q_m/\gamma_2$ , the BPS equations (2.11) give the solutions

$$X' = \frac{q_e}{(\vec{r} - \vec{r}_1)^2 + (\zeta_1^\perp \cdot \vec{x})^2} + \frac{2q_e}{(\vec{r} - \vec{r}_2)^2 + (\zeta_2^\perp \cdot \vec{x})^2} - X_0' \ ,$$

$$Y' = \frac{q_m}{(\vec{r} - \vec{r}_1)^2 + (\zeta_1^\perp \cdot \vec{x})^2} + \frac{q_m}{(\vec{r} - \vec{r}_2)^2 + (\zeta_2^\perp \cdot \vec{x})^2} - Y_0' \ , \tag{4.12}$$

which also give the $\vec{E}, \vec{B}$ components of the field strength tensor. As $\vec{r} \to \infty$ we see that

$$\vec{r} \to \infty, \quad x^4, x^5 \text{ finite}: \qquad X' \sim \frac{3q_e}{|\vec{r}|^2} - X_0' \ , \quad Y' \sim \frac{2q_m}{|\vec{r}|^2} - Y_0' \ , \tag{4.13}$$

and the electric and magnetic field components scale as $(\vec{E}, \vec{B}) \propto (3, 2)$, thereby encoding charge $(3, 2)$. The M5-brane deformation in this asymptotic regime traces a line along $(3, 2)$ in the $(X', Y')$ moduli space when

$$\frac{2}{q_e} X_0' = \frac{3}{q_m} Y_0' \ . \tag{4.14}$$

We impose the boundary conditions

$$\vec{\rho} \to (\vec{r}_1, 0, 0): \quad (X', Y') \to \gamma_1 \left( X_1, \frac{q_m}{q_e} X_1 \right); \quad \vec{\rho} \to (\vec{r}_2, 0, 0): \quad (X', Y') \to \gamma_2 \left( Y_1 \frac{2q_e}{q_m}, Y_1 \right) . \tag{4.15}$$

This fixes the transverse distance between the strings in terms of the scalar moduli values

$$\frac{1}{(\vec{r}_1 - \vec{r}_2)^2} = \frac{1}{2q_m} Y_0' = \frac{1}{3q_e} X_0' \ . \tag{4.16}$$

As in the previous example, we can analyse the $X', Y'$ field configurations in various asymptotic limits, as we go along string$_1$ or string$_2$. This is most efficiently done using the relations (2.11) in terms of the "basis fields" $X, Y$: using (4.11) these become

$$X' = \gamma_1 X + \beta_2 \gamma_2 Y \ , \qquad Y' = \gamma_1 \beta_1 X + \gamma_2 Y \ , \tag{4.17}$$

so

$$X_0' = \gamma_1 X_0 + \beta_2 \gamma_2 Y_0 \ , \qquad Y_0' = \gamma_1 \beta_1 X_0 + \gamma_2 Y_0,$$

$$X_1' = \gamma_1 X_1 \ , \quad Y_1' = \gamma_2 Y_1 \tag{4.18}$$

The resulting scalar asymptotics on the moduli space for $X', Y'$ are tabulated in Table 3 (see Appendix C).

| $(X', Y') \to (X_1', \beta_1 X_1')$ | $\vec{r} \to \vec{r}_1$ | $x_1 = 0$ |
|---|---|---|
| $(X', Y') \to (\, X_1' - (2X_0' - \beta_2 Y_0')\ ,\ \beta_1 X_1' - (\beta_1 X_0' - Y_0')\,)$ | $\vec{r} \to \vec{r}_1$ | $|x_1| \to \infty$ |
| $(X', Y') \to (\beta_2 Y_1'\ ,\ Y_1')$ | $\vec{r} \to \vec{r}_2$ | $x_2 = 0$ |
| $(X', Y') \to (\, \beta_2 Y_1' - (\beta_2 Y_0' - X_0')\ ,\ Y_1' - (2Y_0' - \beta_1 X_0')\,)$ | $\vec{r} \to \vec{r}_2$ | $|x_2| \to \infty$ |
| $(X', Y') \to (-X_0'\ ,\ -Y_0')$ | $\vec{r} \to \infty$ | $|x_1|,\ |x_2| \to \infty$ |

Table 3: $X', Y'$ moduli asymptotics for $(3,2) - (1,1) - (2,1)$

As we approach the wall of marginal stability,

$$X_0',\ Y_0' \ \ll\ X_1',\ Y_1'\,, \tag{4.19}$$

the moduli approach (i) $(X_1', \beta_1 X_1')$ along the entire string$_1$, *i.e.* as $\vec{\rho} \to (\vec{r}_1, x_1, \beta_1 x_1)$, and (ii) $(\beta_2 Y_1', Y_1')$ along the entire string$_2$, *i.e.* as $\vec{\rho} \to (\vec{r}_2, \beta_2 x_2, x_2)$. This is similar to the previous cases, $(1,1) - (1,0) - (0,1)$ and $(2,1) - (1,0) - (1,1)$.

## 4.3   1-string tension

Along the same lines as in Sec. 3.3, we can calculate the tension of the M2-brane web for the configurations we have described. For the $(2,1) - (1,0) - (1,1)$ M2-web, we obtain

$$T_{(2,1)} \ =\ L\left(\,(X_1' \ +\ \beta\, Y_1' \ +\ 2X_0')\, q_e \ +\ (Y_1' + Y_0')\, q_m\,\right), \tag{4.20}$$

while for the $(3,2) - (1,1) - (2,1)$ M2-web, we obtain

$$T_{(3,2)} \ =\ L\left(\,(X_1' \ +\ 2\beta_2\, Y_1' \ +\ 3X_0')\, q_e \ +\ (\beta_1\, X_1' + Y_1' + 2Y_0')\, q_m\,\right). \tag{4.21}$$

These can again be interpreted in terms of the tensions of three M2-brane legs of appropriate charge and shapes, along the lines of (3.49) and (3.50), using the $(m, n)$ membrane tension and the moduli constraints in (4.5) and (4.14) respectively. Various details are described in App. C and specifically App. C.3.

In the formula for energy functional (2.12) there is another term which could give contribution to the tensions in (4.20) and (4.21). The cross term: $(\zeta^1 \cdot \zeta^2)\,(\partial X \cdot \partial Y)$ in (2.12) is non-zero for the examples above with two non-orthogonal strings. In App. C.3, we analyze this term in some detail and discuss the non-trivial contribution that could come from this term. However, in the limit where we consider the transverse distance between the line sources to be large, we find that the contribution from this cross term becomes negligible. Thus near the wall of marginal stability limit (where $X_0, Y_0 \sim 0$) the collective tensions of the configuration in these examples essentially approach those in (4.20) and (4.21), respectively.

# 5 Embedding in "Higgsed" M5-brane theories

We now consider embedding our description of M5-brane prongs into the theory of three M5-branes, parallel and "Higgsed" (*i.e.* on the tensor branch). The schematics with multiple M5s here is similar to that with multiple D3-branes in [19, 21]. The energy functional is

$$\mathcal{E} = \frac{1}{2} \sum_{i=1,2,3} \left( |\partial X^{(i)}|^2 + |\partial Y^{(i)}|^2 + \frac{1}{2} |\tilde{\mathcal{H}}^{(i)}|^2 \right), \tag{5.1}$$

where each M5–brane $_i$ is regarded as independent of the other two. This is essentially three copies of (2.3). This is reasonable in the low energy abelian approximation where the only interaction between the branes arises at the locations where the M2-brane spike legs from one M5-brane join those from another. These locations necessarily include nonabelian contributions, lying outside our approximations: we incorporate their effects through the moduli boundary conditions.

Since the branes are decoupled, we carry out Bogomolny completions (2.5) independently within each brane$_i$ sector, giving (2.6) for each sector. Componentwise these lead via (2.10) to (2.11) for each brane$_i$. Since the M2-brane stretching out from one M5-brane ends on another M5-brane, the line source conditions (2.15) for these must be correlated: likewise the boundary conditions (2.19) are correlated. It is easiest to illustrate this via simple examples.

A simple electrically charged configuration of a single M2-brane stretching between two M5-branes is described by juxtaposing two configurations (3.5). Using the form (3.8), we have

$$X^{(1)} = \frac{q_e}{(\vec{r} - \vec{r}_0^{(1)})^2 + (\zeta^\perp \cdot (\vec{x} - \vec{x}_0^{(1)}))^2} - X_0^{(1)}; \qquad \vec{E}^{(1)} = \nabla X^{(1)}, \quad \Pi^{(1)} = -\partial_5 X^{(1)},$$

$$X^{(2)} = -\frac{q_e}{(\vec{r} - \vec{r}_0^{(2)})^2 + (\zeta^\perp \cdot (\vec{x} - \vec{x}_0^{(2)}))^2} - X_0^{(2)}; \qquad \vec{E}^{(2)} = \nabla X^{(2)}, \quad \Pi^{(2)} = -\partial_5 X^{(2)}. \tag{5.2}$$

In order that the M5-spike$_1$ joins with M5-spike$_2$, we must have

$$\vec{r}_0^{(1)} = \vec{r}_0^{(2)}, \qquad \vec{x}_0^{(1)} = \vec{x}_0^{(2)}. \tag{5.3}$$

The fact that $\vec{E}^{(1)}$ and $\vec{E}^{(2)}$ are opposite in sign is tantamount to the statement that the M2-brane stretching between the two separated M5-branes describes a charge $(1,0)_1 - (-1,0)_2$ from the point of view of the $U(1)_1 \times U(1)_2$ theory, "Higgsed" from the $U(2)$ theory of two M5-branes.

Now in addition, the fact that the two spike "cores" join at the same location in moduli space requires that they be the same "core size" within this abelian approximation: using

(3.7), this gives

$$q_e\alpha_{00}^{(1)} - X_0^{(1)} \sim X_1^{(1)} = X_1^{(2)} \sim -q_e\alpha_{00}^{(2)} - X_0^{(2)} \ ,$$

$$\alpha_{00}^{(1)} = \alpha_{00}^{(2)} \qquad \Rightarrow \qquad X_0^{(1)} + X_1^{(1)} = -X_1^{(1)} - X_0^{(2)} \ ,$$

$$\Rightarrow \qquad X_1^{(1)} = -\frac{1}{2}(X_0^{(1)} + X_0^{(2)}) \ , \qquad \alpha_{00}^{(1)} = \alpha_{00}^{(2)} = \frac{1}{2q_e}(X_0^{(1)} - X_0^{(2)}) \ . \tag{5.4}$$

The core size thus scales as the inverse distance between the M5-branes which are located at $-X_0^{(1)}$ and $-X_0^{(2)}$ (where the deformations in (5.2) vanish). Also, we see that the gluing location in moduli space is at the midpoint between the two M5-branes. This is the location of enhanced $U(2)$ symmetry in moduli space where new light nonabelian tensor modes must be added to the low energy theory for a nonsingular description.

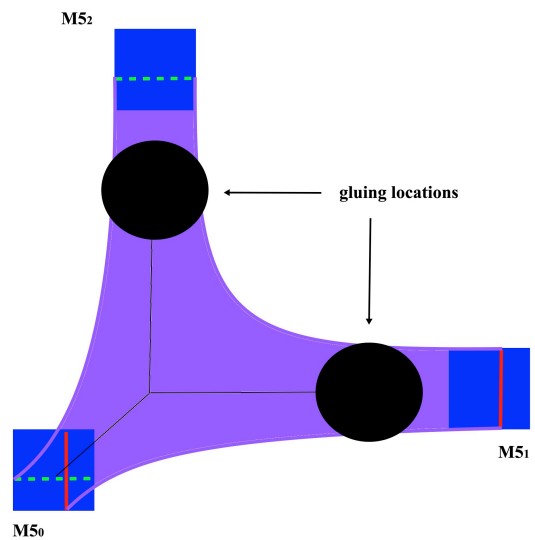

Figure 2: Cartoon of three parallel separated M5-branes with a prong deformation stretching between them in the transverse $(X, Y)$-space. The solid red and dashed green lines depict the non-parallel M2-endlines in the M5-brane worldvolumes (spanning $\{0, 1, 2, 3, 4, 5\}$). The solid black circles represent the gluing locations, where the prong legs are joined to the corresponding spike legs.

Along similar lines, we can embed the simplest M5-brane prong configurations in a theory of three parallel separated M5-branes located as in Figure 2 (drawing this accurately requires more dimensions than possible(!); the directions are as in (3.10)), which completes Fig. 1 (d):

$$M5_0: \ (-X_0^{(0)}, -Y_0^{(0)}); \qquad M5_1: \ (X_0^{(1)}, 0); \qquad M5_2: \ (0, Y_0^{(2)});$$

$$X_0^{(0)}, Y_0^{(0)} \ll X_0^{(1)}, \ Y_0^{(2)} \ . \tag{5.5}$$

The last condition encodes the fact that M5-brane $_0$ is near the wall of marginal stability so its $X, Y$ scalars are turned on in a prong-like field configuration (3.12). Each constituent spike leg of this prong is glued onto a corresponding spike (3.5) from one of the other M5-branes.

The field configurations then are (with $Y^{(1)} = 0$ and $X^{(2)} = 0$)

$$\vec{E}^{(0)} = \nabla X^{(0)} , \qquad X^{(0)} = \frac{q_e}{(\vec{r} - \vec{r}_1^{(0)})^2 + (\zeta_1^{\perp} \cdot (\vec{x} - \vec{x}_1^{(0)}))^2} - X_0^{(0)} ,$$

$$\vec{B}^{(0)} = \nabla Y^{(0)} , \qquad Y^{(0)} = \frac{q_m}{(\vec{r} - \vec{r}_2^{(0)})^2 + (\zeta_2^{\perp} \cdot (\vec{x} - \vec{x}_2^{(0)}))^2} - Y_0^{(0)} ,$$

$$\Pi^{(0)} = \partial_4 Y^{(0)} - \partial_5 X^{(0)} , \qquad\qquad\qquad\qquad\qquad\qquad (5.6)$$

$$\vec{E}^{(1)} = \nabla X^{(1)} , \quad X^{(1)} = -\frac{q_e}{(\vec{r} - \vec{r}_0^{(1)})^2 + (\zeta_1^{\perp} \cdot (\vec{x} - \vec{x}_0^{(1)}))^2} + X_0^{(1)} , \qquad \Pi^{(1)} = -\partial_5 X^{(1)} ,$$

$$\vec{B}^{(2)} = \nabla Y^{(2)} , \quad Y^{(2)} = -\frac{q_m}{(\vec{r} - \vec{r}_0^{(2)})^2 + (\zeta_2^{\perp} \cdot (\vec{x} - \vec{x}_0^{(2)}))^2} + Y_0^{(2)} , \qquad \Pi^{(2)} = \partial_4 Y^{(2)} .$$

The fact that the $\vec{E}^{(0)}, \vec{E}^{(1)}$ are of opposite sign (and likewise $\vec{B}^{(0)}, \vec{B}^{(2)}$) indicates the fact that the M2-branes stretching between the three parallel separated M5-branes describes a charge $(1,1)_0 - (-1,0)_1 - (0,-1)_2$ configuration from the point of view of the $U(1)_0 \times U(1)_1 \times U(1)_2$ theory, "Higgsed" from the $U(3)$ theory of three M5-branes.

In order that the prong leg-$X^{(0)}$ joins with spike$_1$ and correspondingly prong leg-$Y^{(0)}$ joins with spike$_2$, we must have

$$\vec{r}_1^{(0)} = \vec{r}_0^{(1)} , \quad \vec{x}_1^{(0)} = \vec{x}_0^{(1)} ; \qquad\qquad \vec{r}_2^{(0)} = \vec{r}_0^{(2)} , \quad \vec{x}_2^{(0)} = \vec{x}_0^{(2)} , \qquad\qquad (5.7)$$

*i.e.* the string locations in the various M5-branes must match, or equivalently the M2-brane stretching between M5-brane$_0$ and M5-brane$_1$ (and likewise M5-brane$_0$ and M5-brane$_2$) creates endlines whose worldvolume locations must match.

In addition, the string "core sizes" must match between M5-brane$_0$ and M5-brane$_1$ (and likewise M5-brane$_0$ and M5-brane$_2$). Applying (3.23) and (3.7) to the M5-branes appropriately, this matching requires

$$q_e \alpha_{11}^{(0)} - X_0^{(0)} \sim X_1^{(0)} = X_1^{(1)} \sim -q_e \alpha_{00}^{(1)} + X_0^{(1)} , \qquad \alpha_{11}^{(0)} = \alpha_{00}^{(1)} ;$$

$$q_m \alpha_{22}^{(0)} - Y_0^{(0)} \sim Y_1^{(0)} = Y_1^{(2)} \sim -q_m \alpha_{00}^{(2)} + Y_0^{(2)} , \qquad \alpha_{22}^{(0)} = \alpha_{00}^{(2)} . \qquad (5.8)$$

Solving for the moduli locations $(X_1^{(0)}, 0)$ and $(0, Y_1^{(0)})$ at which we perform the gluing and the inverse core sizes $\alpha_{ii}$, we obtain

$$X_0^{(0)} + X_1^{(0)} = -X_1^{(0)} + X_0^{(1)} ; \qquad Y_0^{(0)} + Y_1^{(0)} = -Y_1^{(0)} + Y_0^{(2)} ,$$

$$\Rightarrow \qquad X_1^{(0)} = \frac{1}{2}(X_0^{(1)} - X_0^{(0)}) , \qquad \alpha_{11}^{(0)} = \alpha_{00}^{(1)} = \frac{1}{2q_e}(X_0^{(0)} + X_0^{(1)}) ,$$

$$\Rightarrow \qquad Y_1^{(0)} = \frac{1}{2}(Y_0^{(2)} - Y_0^{(0)}) , \qquad \alpha_{22}^{(0)} = \alpha_{00}^{(2)} = \frac{1}{2q_m}(Y_0^{(0)} + Y_0^{(2)}) . \qquad (5.9)$$

Using the locations (5.5) of the three M5-branes, we see that the gluing occurs at the midpoint between M5-brane$_0$ and M5-brane$_1$ (and likewise M5-brane$_0$ and M5-brane$_2$). These are

the locations of enhanced gauge symmetry where the stretched M2-branes become massless signalling light nonabelian degrees of freedom. The inverse core sizes scale as the distance between the corresponding pairs of M5-branes. Finally, analysing the moduli values in the vicinity of the string soliton cores as in *e.g.* (3.20) suggests the boundary conditions

$$\vec{\rho} \to (\vec{r}_1^{(0)}, 0, 0): \qquad (X^{(0)}, Y^{(0)}) \to (X_1^{(0)}, 0) \ \leftarrow (X^{(1)}, Y^{(1)}) \ ,$$
$$\vec{\rho} \to (\vec{r}_2^{(0)}, 0, 0): \qquad (X^{(0)}, Y^{(0)}) \to (0, Y_1^{(0)}) \ \leftarrow (X^{(2)}, Y^{(2)}) \ , \qquad (5.10)$$

analogous to (3.22). This leads to the transverse separation between the string solitons on M5$_0$ as $(\vec{r}_1^{(0)} - \vec{r}_2^{(0)})^2 = \frac{q_e}{X_0^{(0)}}$, analogous to (3.21), with a similar wall-crossing limit $X_0^{(0)} \to 0$.

# 6   Discussion

We have constructed and described field configurations representing self-dual string soliton bound states in the M5-brane abelian effective field theory. These resemble M5-brane "prongs" as we have seen: the simplest such configuration is depicted in Figure 1 (to the extent possible: a more full picture requires more dimensions than can be drawn!). While the Bogomolny completion (2.5) and the resulting BPS bound equations (2.6) appear general and written in terms of intrinsically M5-brane structures, the explicit field configurations are best expressed and understood in terms of the component forms (2.10), (2.11) and (2.12), which decompose the self-dual 3-form tensor field strength (2.2) into appropriate electric and magnetic components (2.9). It may be interesting to understand how our starting point, the effective low energy functional (2.3), gels with the recent action formulations of self-dual forms [40, 41] (see also [42]).

A single string soliton representing the endline of an M2-brane ending on the M5-brane is described by an M5-brane spike deformation (3.5) sourced by a single scalar field: an electric string has field strength components $\tilde{H}^{a4}$, $\tilde{H}^{45}$ ($a = 1, 2, 3$). Bound states of two string solitons described by M5-brane prong deformations, have nonvanishing field strength components $\tilde{H}^{a4}$, $\tilde{H}^{a5}$, $\tilde{H}^{45}$, sourced by two scalar fields. To describe bound states of string solitons, operationally, we wrote down solutions to the BPS equations that describe a superposition of two non-parallel string solitons (the 6-dim theory has only self-dual strings so "electric/magnetic" simply label the 4/5-directions that the strings are stretched along; the non-parallelness of the strings leads to electric/magnetic in 4-dimensions upon 45-compactification). In the simplest case of two orthogonal strings, these are (3.12), with behaviour as described subsequently. We then impose boundary conditions describing the scalar moduli values as we approach the string locations: these are imposed at the closest point of approach of the two strings, given by (3.22) in the simplest case (analogous to (3.6)

for a single string but with more features). This is equivalent to fixing the separation between the strings transverse to their extended directions in terms of the distance from the wall of marginal stability, as in (3.21). As we have seen, imposing these boundary conditions or more generally (2.19) at the closest approach becomes an increasingly better approximation to imposing them all along the strings (*i.e.* at $\vec{\rho}_{\zeta^1} \to \vec{\rho}^{\,0}_{\zeta^1}$ and $\vec{\rho}_{\zeta^2} \to \vec{\rho}^{\,0}_{\zeta^2}$), in the limit of approaching the wall of marginal stability in moduli space, as we discussed in detail around (3.22). In this limit, the separation between the solitons increases and eventually diverges (3.24), so the state then becomes unbound (and disappears from the spectrum as one crosses the wall). While there are clear parallels with string web states in super Yang-Mills theories, it may be of importance to understand wall-crossing phenomena more systematically for such string-string bound state molecules and the role played by the extended nature of the constituents, for instance in regard to their degeneracies which are likely to have more features compared with [43, 44] (more generally see *e.g.* [45, 46] for discussions on wall-crossing in relation to black hole microstate counting).

Our description of these bound states has been indirect in a sense: we have used the BPS bound equations and imposed appropriate boundary conditions whose intuition stems from the M5-M2-M2 brane geometry we expect in the space transverse to the M5-brane (and corresponding descriptions of string web states via D3-brane deformations [19, 21]). Our equilibrium configurations (reliable in the limit of approaching the wall of marginal stability) suggest that the scalar forces balance the 3-form forces between the constituent string solitons. It would be most interesting to understand the nature of these bound states more intrinsically in the M5-brane tensor theory, *e.g.* in terms of the way the 3-form field lines behave for such composite configurations and the corresponding forces, more directly, between the constituent string solitons.

A related point is the following. As we have seen, we have imposed boundary conditions (3.22) (and more generally (2.19)) at the point of closest approach of the two strings: this leads to the moduli variations and associated brane-bendings, as in Table 1, which become refined as we approach marginal stability. One might ask if, instead, one could impose boundary conditions that amount to holding the strings fixed far along them (at infinity), and allowing the rest of the two-string geometry (at finite distances) be determined by energetics. It would seem that in this case the scalar configurations will be different from (3.12), leading to brane-bendings that reflect in the two strings deformed towards each other in their near regions. This might possibly be a different energy extremization than (2.5) with different boundary terms held fixed (see sec. 3.4 for some comments on strict Bogomolny completions versus the extra terms in (2.5)). In a sense this might be more natural if we consider the dynamical problem of a self-dual string probe moving in the background field of another non-parallel self-dual string: the forces will be less severe far along the string probe

(at infinity) while the string bulk will deform more. In the limit of approaching marginal stability, from our discussions of the field configurations, moduli variations and the tension in sec. 3.3 and sec. 3.4, it is clear that the above description will approach our formulation. In some sense this sort of string-string dynamics might be describable by appropriate string uplifts of the particle dynamics in [31], [32]. It would be instructive to understand these issues better, and we hope to do so.

Perhaps understanding these string-string effective potentials could be explored building on [47] (see *e.g.* [48, 49] for some interesting related discussions). One might hope thereby to gain valuable insight into the dynamics of self-dual string solitons and thence that of self-dual 3-forms in the nonabelian M5-brane theory (see *e.g.* [50, 51] for nonabelian self-dual strings, which may bear on our discussions in sec. 5 on embedding into multiple abelian M5-brane theories). Relatedly, it would be interesting to understand more systematically the role of supersymmetry with regard to our Bogomolny completion, appropriately generalizing [52]: this may dovetail with the role of fermionic zero modes in the effective supersymmetric sigma model for small string-string fluctuations.

It is worth noting that our Bogomolny completion is in a sense tied to the existence of two $\zeta$-vectors in the $(x^4, x^5)$-plane: this means the two M2-branes ending on the M5-brane stretch in this 2-dim subspace alone so the 3-dim $\vec{r}$-subspace is untouched. In some essential sense, this dovetails with the intuition that the these M5-M2-M2 configurations will compactify to become appropriate D3-brane string-web configurations with the $\vec{r}$-subspace being the D3-brane worldvolume. On the other hand, one could imagine more general configurations such as with three or more M2-branes ending on the M5-brane (see [23] for some investigations on these). These may relate to the D3-brane prong configurations in [19, 21] pertaining to three or more charge centers (see *e.g.* [53] for multiple center black hole configurations). It would be interesting to explore these further.

There are also a few other interesting questions that come to mind. One pertains to generalizing these M5-brane prong configurations to less supersymmetric theories such as those in *e.g.* [54] and [55], possibly interlinking with [39], [56]: these might be related to D3-brane prong descriptions [19] of string webs [57, 58, 59] in $\mathcal{N}=2$ SYM theories [60], and might exhibit appropriate s-rules in regard to the BPS spectrum. Another pertains to whether these sorts of string soliton bound states in the 6-dim $(2,0)$ theory can be realized explicitly as field configurations in the nonperturbative 5-dim SYM theory description discussed in [61, 62]. A third involves understanding 5-brane webs *e.g.* [63] (see also [64]) as connected prong-like brane-deformations. A still further question has to do with whether an M2-brane web (regarded as an object in itself, with no M5-brane) can be described from the point of view of the M2-brane worldvolume theory itself [65, 66, 67, 68]: in some sense this might be analogous to the description of a string web (in the absence of a D3-brane) from the point

of view of the D-string worldvolume gauge theory [69]. Perhaps more general M2-brane networks in M-theory might be related to IIB string networks [70].

Our analysis of the M5-brane prong configurations may also be useful in understanding other BPS observables in the 6d worldvolume theory. The analysis of surface observables in [71], [72] uses a similar M2-M5 brane construction to study various geometric features and non-perturbative aspects associated with them in the 6d $(2,0)$ theory. In this regard, our single M5-brane spike configuration is similar to the half-BPS surface observable in these references. But while those works involved the use of $AdS_7 \times S^4$ holography [71] or used geometric constructions on special manifolds [72], a detailed analysis in the theory on a single M5 worldvolume is not done in this way. It would be interesting to see if our configurations with the M5-prong structures help uncover more about these BPS surface observables.

**Acknowledgements:** It is a pleasure to thank Philip Argyres, Sujay Ashok, Jerome Gauntlett and Costis Papageorgakis for helpful comments on a draft. This work is partially supported by a grant to CMI from the Infosys Foundation.

# A    Useful identities

This appendix contains some details on intermediate steps involved in obtaining the Bogomolnyi completion in (2.5) from the energy functional expression (2.3).

- First we have

$$\left| \partial_{[a} X \zeta_{b]}^{(1)} \right|^2 = \sum_{a \neq b\,;\,c \neq d} \frac{\delta^{ac}\delta^{bd}}{4} \left( \partial_a X \zeta_b^{(1)} - \partial_b X \zeta_a^{(1)} \right) \left( \partial_c X \zeta_d^{(1)} - \partial_d X \zeta_c^{(1)} \right)$$
$$= \frac{1}{2} \sum_{a \neq b} (\partial_a X)^2 \left( \zeta_b^{(1)} \right)^2 - \frac{1}{2} \sum_{a \neq b} \left( \partial_a X \zeta_a^{(1)} \right) \left( \partial_b X \zeta_b^{(1)} \right) \tag{A.1}$$

- The expansion of $\left( \partial_a X \zeta_a^{(1)} \right)^2$ is the following

$$\sum_{a,b} \left( \partial_a X \zeta_a^{(1)} \right) \left( \partial_b X \zeta_b^{(1)} \right) = \sum_a (\partial_a X)^2 \left( \zeta_a^{(1)} \right)^2 + \sum_{a \neq b} \left( \partial_a X \zeta_a^{(1)} \right) \left( \partial_b X \zeta_b^{(1)} \right) \tag{A.2}$$

And the sum $\left| \partial_{[a} X \zeta_{b]}^{(1)} \right|^2 + \frac{1}{2} \times \left( \partial_a X \zeta_a^{(1)} \right)^2$ is equal to $\frac{1}{2} |\partial X|^2$.

- The dot product $(\partial_{[a} X \zeta_{b]}^{(1)}) \cdot (\partial_{[a} Y \zeta_{b]}^{(2)})$ can be expanded as

$$\partial_{[a} X \zeta_{b]}^{(1)} \partial_{[c} Y \zeta_{d]}^{(2)} \delta^{ac} \delta^{bd} = \sum_{a \neq b\,;\,c \neq d} \frac{\delta^{ac}\delta^{bd}}{4} \left( \partial_a X \zeta_b^{(1)} - \partial_b X \zeta_a^{(1)} \right) \left( \partial_c Y \zeta_d^{(2)} - \partial_d Y \zeta_c^{(2)} \right)$$
$$= \frac{1}{2} \sum_{a \neq b} (\partial_a X \partial_a Y) \left( \zeta_b^{(1)} \zeta_b^{(2)} \right) - \frac{1}{2} \sum_{a \neq b} \left( \partial_a X \zeta_a^{(2)} \right) \left( \partial_b Y \zeta_b^{(1)} \right) \tag{A.3}$$

- And the expansion of $\left(\partial_a X \zeta_a^{(1)}\right)\left(\partial_b Y \zeta_b^{(2)}\right)$ is the following

$$\sum_{a,b} \left(\partial_a X \zeta_a^{(1)}\right)\left(\partial_b Y \zeta_b^{(2)}\right) = \sum_a \left(\partial_a X \partial_a Y\right)\left(\zeta_a^{(1)} \zeta_a^{(2)}\right) + \sum_{a \neq b} \left(\partial_a X \zeta_a^{(1)}\right)\left(\partial_b Y \zeta_b^{(2)}\right) .$$
(A.4)

So the addition: $2 \times (\partial_{[a} X \zeta_{b]}^{(1)}) \cdot (\partial_{[a} Y \zeta_{b]}^{(2)}) + \left(\partial_a X \zeta_a^{(1)}\right)\left(\partial_b Y \zeta_b^{(2)}\right)$ gives the following answer

$$\sum_{a,b} \left(\partial_a X \partial_a Y\right)\left(\zeta_b^{(1)} \zeta_b^{(2)}\right) + \sum_{a \neq b} \zeta_a^{(1)} \zeta_b^{(2)} \left(\partial_a X \partial_b Y - \partial_b X \partial_a Y\right) .$$
(A.5)

# B  A brief review: D3-brane prongs and string webs

Here we give a short recap of the D3-brane prong description of string web states in [19, 20, 21]. The low energy abelian effective action on a D3-brane leads to an energy functional

$$\mathcal{E} = \frac{1}{2}\left(\vec{E}^2 + \vec{B}^2 + (\nabla X)^2 + (\nabla Y)^2\right) \;\rightarrow\; \frac{1}{2}\left[(\vec{E} - \nabla X')^2 + (\vec{B} - \nabla Y')^2\right] + \vec{E}\cdot\nabla X' + \vec{B}\cdot\nabla Y'$$
(B.1)

where $X' = X \cos\phi + Y \sin\phi$ and $Y' = -X \sin\phi + Y \cos\phi$. The angle $\phi$ can effectively be set to zero analogous to (3.47): this amounts to an overall rotation of $X, Y$, which leads to an analog of the moduli constraint $e.g.$ (3.16) mapping the charge to the asymptotic shape of the D3-brane deformation. We thus obtain the BPS bound equations

$$\vec{E} = \nabla X , \qquad \vec{B} = \nabla Y ; \qquad \nabla \cdot \vec{E} = \nabla^2 X = 0 , \quad \nabla \cdot \vec{B} = \nabla^2 Y = 0 .$$
(B.2)

so the scalars $X, Y$ are harmonic in the D3-brane spatial worldvolume. The rotation to $\phi = 0$ implies that electric charge configurations are sourced by nontrivial $X$ scalar deformations of the D3-brane worldvolume (in the transverse $X$-direction) while magnetic ones are sourced by nontrivial $Y$ scalar D3-brane deformations. Generic solutions to these represent string web states in the gauge theory. A simple example is the field configuration

$$X = \frac{q_e}{|\vec{r} - \vec{r}_1|} - X_0 , \qquad Y = \frac{q_m}{|\vec{r} - \vec{r}_2|} - Y_0 ,$$
(B.3)

representing a charge $(1,1) - (1,0) - (0,1)$ state, with one electric and one magnetic charge center. Its asymptotic behaviour is

$$|\vec{r}| \to \infty : \qquad X \sim \frac{q_e}{|\vec{r}|} - X_0 , \quad Y \sim \frac{q_m}{|\vec{r}|} - Y_0 ,$$
(B.4)

so $(\vec{E}, \vec{B}) \propto (1,1)$ encodes charge $(1,1)$ in $q_e, q_m$ units. This deformation traces a line along $(1,1)$ in the moduli space if $\frac{1}{q_e}X_0 = \frac{1}{q_m}Y_0$. Further as we approach either charge core, we see

$$\vec{r} \to \vec{r}_1 : \quad Y \sim \frac{q_m}{|\vec{r}_1 - \vec{r}_2|} - Y_0 ; \qquad \vec{r} \to \vec{r}_2 : \quad X \sim \frac{q_e}{|\vec{r}_1 - \vec{r}_2|} - X_0 ,$$
(B.5)

while $X \to X_1$ and $Y \to Y_1$ develop long spikes respectively (which we have cut off). It is then consistent to impose the boundary conditions on the moduli as

$$\vec{r} \to \vec{r}_1 : \quad (X, Y) \to (X_1, 0) \,; \qquad \vec{r} \to \vec{r}_2 : \quad (X, Y) \to (0, Y_1) \,. \tag{B.6}$$

This leads to fixing the separation between the charge cores

$$\frac{1}{|\vec{r}_1 - \vec{r}_2|} \sim \frac{1}{q_m} Y_0 = \frac{1}{q_e} X_0 \,, \tag{B.7}$$

which is large relative to the core sizes (which scale as $\frac{1}{X_1}$ and $\frac{1}{Y_1}$) in the limit of approaching the wall of marginal stability, with $X_0, Y_0 \ll X_1, Y_1$. As we hit this wall, $i.e.$ $X_0, Y_0 \to 0$, the state becomes arbitrarily loosely bound and decays. No bound state solutions exist on the other side of the wall (for $X_0 < 0$).

More general field configurations can be written as solutions to the BPS bound equations, with similar reliable behaviour near the wall of marginal stability.

# C   Some details on more general M5-M2 states

In this appendix we give some details of use in analyzing the brane deformations presented in the main section (4). The expression for the "basis fields" $X$ and $Y$ are the following

$$X = \frac{c_X}{(\vec{r} - \vec{r}_1)^2 + (\zeta_1^\perp \cdot \vec{x})^2} - X_0 \,, \qquad Y = \frac{c_Y}{(\vec{r} - \vec{r}_2)^2 + (\zeta_2^\perp \cdot \vec{x})^2} - Y_0 \tag{C.1}$$

## C.1   (2,1) - (1,0) - (1,1)

For the $(2, 1) - (1, 0) - (1, 1)$ M5-prongs we consider $X, Y$ with $c_X = q_e$ and $c_Y = q_m/\gamma$. At large radial distance from the string sources $|\vec{r}| \to \infty$ these "basis" fields approach

$$|\vec{r}| \to \infty : \quad X \sim -X_0 \qquad Y \sim -Y_0 \,. \tag{C.2}$$

As we approach string$_1$, with coordinates in (4.1a), the $X$ field develops a spike while the $Y$ field is nonsingular:

$$\vec{\rho} \to (\vec{r}_1, x_1, 0) : \qquad X \to X_1 \qquad Y \sim \frac{c_Y}{(\vec{r}_1 - \vec{r}_2)^2 + \gamma^2 (x_1)^2} - Y_0 \tag{C.3}$$

Analogous to the $(1, 1) - (1, 0) - (0, 1)$ example, at $(r_1, 0, 0)$ we impose

$$Y = 0 \quad \implies \quad \frac{c_Y}{(\vec{r}_1 - \vec{r}_2)^2} = Y_0 \tag{C.4}$$

On approaching string$_2$ (onto (4.1b)) the $Y$ field develops a spike while $X$ is nonsingular:

$$\vec{\rho} \to (r_2, \beta x_2, x_2) : \qquad X \sim \frac{q_e}{(\vec{r}_2 - \vec{r}_1)^2 + (x_2)^2} - X_0 \qquad Y \to Y_1 \,. \tag{C.5}$$

So at $(r_2, 0, 0)$, we impose

$$X = 0 \quad \Longrightarrow \quad \frac{q_e}{(\vec{r}_2 - \vec{r}_1)^2} = X_0 \tag{C.6}$$

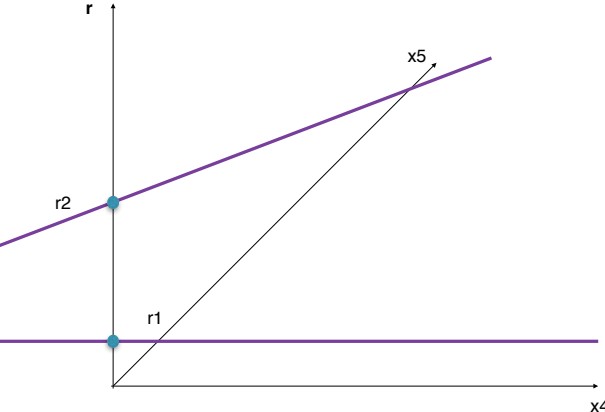

Figure 3: This depicts the orientations and the positions of the two strings in the 5-dimensional space on the worldvolume. The two strings are not orthogonal when projected on the $x^4$-$x^5$ plane.

Here we see that the values of $X$ and $Y$ fields interpolate as we move along the two strings away from the origin. These values are tabulated in Table 1, except they are to be interpreted as values for the "basis" fields for our purposes now. Using these asymptotics for $X, Y$ and the relations (4.8), we obtain the asymptotics for $X', Y'$ in Table 2.

## C.2 (3,2) - (1,1) - (2,1)

For the $(3, 2) - (1, 1) - (2, 1)$ prongs we consider $X$, $Y$ in (C.1) with $c_X = q_e/\gamma_1$ and $c_Y = q_m/\gamma_2$.

When the radial distance is large from the string sources $|\vec{r}| \to \infty$ these "basis" fields approach $(-X_0, -Y_0)$ as before. As in the previous example as we approach string$_1$ with coordinates in (4.11a)), the $X$ field develops a spike and the $Y$ field is nonsingular:

$$\vec{\rho} \to (\vec{r}_1, x_1, \beta_1 x_1) : \qquad X \to X_1 \qquad Y \sim \frac{c_Y}{(\vec{r}_1 - \vec{r}_2)^2 + \gamma_2^2 (x_1)^2} - Y_0 \tag{C.7}$$

Analogous to the previous example, at $(r_1, 0, 0)$ we impose

$$Y = 0 \quad \Longrightarrow \quad \frac{c_Y}{(\vec{r}_1 - \vec{r}_2)^2} = Y_0 \tag{C.8}$$

On approaching string$_2$ (see (4.11b)) there is a spike in the value of the $Y$ field while the $X$ field is nonsingular:

$$\vec{\rho} \to (r_2\,,\,\beta_2\,x_2\,,\,x_2)\,: \qquad X \sim \frac{c_X}{(\vec{r_2} - \vec{r_1})^2 + \gamma_1^2(x_2)^2} - X_0 \qquad Y \to Y_1 \qquad \text{(C.9)}$$

So at $(r_2, 0, 0)$, we impose

$$X = 0 \quad \implies \quad \frac{c_X}{\left(\vec{r_2} - \vec{r_1}\right)^2} = X_0 \qquad \text{(C.10)}$$

Again we see that the values of $X$ and $Y$ fields interpolate as we move along the two strings away from the origin. Although the detailed values of $X, Y$ change in the regions along the strings, in the asymptotic regions they become the same as in Table 1. Using these and the relations (4.17), we obtain the asymptotics for $X', Y'$ in Table 3.

## C.3   1-string tension

First we consider the $(2, 1)-(1, 0)-(1, 1)$ web. The boundary terms in (2.12) upon integrating become

$$\int dx^4 dx^5 \int d^3r \; \tilde{\mathcal{H}}^{a4} \partial_a X' + \int dx^4 dx^5 \int d^3r \; \tilde{\mathcal{H}}^{a5} \, \partial_a Y' \qquad \text{(C.11)}$$

In terms of $X, Y$ fields this is

$$\int dx^4 dx^5 \int d^3r \; \tilde{\mathcal{H}}^{a4} \left(\partial_a X \,+\, \beta\gamma \, \partial_a Y\right) + \int dx^4 dx^5 \int d^3r \; \tilde{\mathcal{H}}^{a5} \, \gamma \, \partial_a Y \qquad \text{(C.12)}$$

For convenience we write down the explicit expression for these fields

$$X = \frac{q_e}{(\vec{r} - \vec{r_1})^2 + (x^5)^2} - X_0\,, \qquad Y = \frac{q_m/\gamma}{(\vec{r} - \vec{r_2})^2 + \gamma^2(x^4 - \beta x^5)^2} - Y_0 \qquad \text{(C.13)}$$

In (C.12) we do the integration by parts and subsequently use the equation of motion $\partial_a \tilde{\mathcal{H}}^{ab} = 0$ to get

$$\int dx^4 \oint_{S^3} d\hat{s}^a \; \tilde{\mathcal{H}}^{a4} \, X + \beta\gamma \int d\tilde{x}^5 \oint_{S^3} d\hat{s}^a \; \tilde{\mathcal{H}}^{a4} \, Y + \gamma \int d\tilde{x}^5 \oint_{S^3} d\hat{s}^a \; \tilde{\mathcal{H}}^{a5} \, Y \qquad \text{(C.14)}$$

where $d\tilde{x}^5$ is the differential length element along string$_2$ and should be expressible as: $d\tilde{x}^5 = \gamma \, (\beta dx^4 + dx^5)$. We are considering two strings with charges

$$(Q_E^i, Q_B^i), \; i = 1, 2\,: \qquad Q_E^i = \oint_{S_i^3} \tilde{\mathcal{H}}^{a4} ds^a\,, \qquad Q_B^i = \oint_{S_i^3} \tilde{\mathcal{H}}^{a5} ds^a\,, \qquad \text{(C.15)}$$

and scalar moduli values at their cores as given in Table (1). At spatial infinity far from both strings, charge conservation gives $(Q_E^0, Q_B^0) = -\sum_i (Q_E^i, Q_B^i)$ and the moduli values

are $(-X_0, -Y_0)$.

$$\int dx^4 \oint_{S^3_{i=1}} d\hat{s}^a \, \tilde{\mathcal{H}}^{a4} X \; + \; \beta\gamma \int d\tilde{x}^5 \oint_{S^3_{i=2}} d\hat{s}^a \, \tilde{\mathcal{H}}^{a4} Y \; + \; \gamma \int d\tilde{x}^5 \oint_{S^3_{i=2}} d\hat{s}^a \, \tilde{\mathcal{H}}^{a5} Y$$

$$+ \int dx^4 \oint_{S^3_\infty} d\hat{s}^a \, \tilde{\mathcal{H}}^{a4} X \; + \; \beta\gamma \int d\tilde{x}^5 \oint_{\tilde{S}^3_\infty} d\hat{s}^a \, \tilde{\mathcal{H}}^{a4} Y \; + \; \gamma \int d\tilde{x}^5 \oint_{\tilde{S}^3_\infty} d\hat{s}^a \, \tilde{\mathcal{H}}^{a5} Y$$

$$\text{(C.16)}$$

Integrating over 5-space, and so at appropriate $S^3_i$, gives

$$L\left( (X_1 + X_0 + \beta\gamma\, Y_0)\, Q^1_E \; + \; \beta\gamma \left( Y_1 + Y_0 + \frac{X_0}{\beta\gamma} \right) Q^2_E \; + \; \gamma(Y_1 + Y_0) Q^2_B \right), \qquad \text{(C.17)}$$

with $L = \int dx^4 \sim \int d\tilde{x}^5$ the regulated length of both strings. Since $(Q^1_E, Q^1_B) = (q_e, 0)$ and $(Q^2_E, Q^2_B) = (q_e, q_m)$ this is equal to

$$L\left( (X_1 + X_0 + \beta\gamma\, Y_0)\, q_e \; + \; \beta\gamma \left( Y_1 + Y_0 + \frac{X_0}{\beta\gamma} \right) q_e \; + \; \gamma(Y_1 + Y_0) q_m \right) \qquad \text{(C.18)}$$

Further, the above formula can also be written in terms of $X'$ and $Y'$ field values as given in (4.20) in section 4.3.

Likewise, for the $(3, 2) - (1, 1) - (2, 1)$ web, the boundary terms upon integrating become

$$\int dx^4 dx^5 \int d^3r \, \tilde{\mathcal{H}}^{a4} \partial_a X' \; + \; \int dx^4 dx^5 \int d^3r \, \tilde{\mathcal{H}}^{a5} \, \partial_a Y' \qquad \text{(C.19)}$$

In terms of $X, Y$ fields this is

$$\int dx^4 dx^5 \int d^3r \, \tilde{\mathcal{H}}^{a4} \left( \gamma_1 \partial_a X \; + \; \beta_2 \gamma_2 \, \partial_a Y \right) \; + \; \int dx^4 dx^5 \int d^3r \, \tilde{\mathcal{H}}^{a5} \left( \beta_1 \gamma_1 \, \partial_a X \; + \; \gamma_2 \, \partial_a Y \right)$$

$$\text{(C.20)}$$

For convenience we write down the explicit expression for these fields

$$X = \frac{q_e/\gamma_1}{(\vec{r} - \vec{r_1})^2 + \gamma_1 (\beta_1 x^4 - x^5)^2} - X_0 \,, \qquad Y = \frac{q_m/\gamma_2}{(\vec{r} - \vec{r_2})^2 + \gamma_2^2 (x^4 - \beta_2 x^5)^2} - Y_0$$

$$\text{(C.21)}$$

In (C.12) we do the integration by parts and subsequently use the equation of motion $\partial_a \tilde{\mathcal{H}}^{ab} = 0$ to get

$$\gamma_1 \int d\tilde{x}^4 \oint_{S^3} d\hat{s}^a \, \tilde{\mathcal{H}}^{a4} X \; + \; \beta_2 \gamma_2 \int d\tilde{x}^5 \oint_{S^3} d\hat{s}^a \, \tilde{\mathcal{H}}^{a4} Y \; + \; \beta_1 \gamma_1 \int d\tilde{x}^4 \oint_{S^3} d\hat{s}^a \, \tilde{\mathcal{H}}^{a5} X$$

$$+ \; \gamma_2 \int d\tilde{x}^5 \oint_{S^3} d\hat{s}^a \, \tilde{\mathcal{H}}^{a5} Y \qquad \text{(C.22)}$$

where $d\tilde{x}^4$ and $d\tilde{x}^5$ are the differential length elements along string$_1$ and string$_2$ respectively, and should be expressible as:

$$d\tilde{x}^4 = \gamma_1 \left( dx^4 + \beta_1 \, dx^5 \right), \qquad d\tilde{x}^5 = \gamma_2 \left( \beta_2 \, dx^4 + dx^5 \right).$$

As before, we are considering two string charges $(Q_E^i, Q_B^i)$, $i = 1, 2$, with scalar moduli values at their cores as given in Table (1). At spatial infinity far from both strings, charge conservation gives $(Q_E^0, Q_B^0) = -\sum_i (Q_E^i, Q_B^i)$. We obtain

$$
\begin{aligned}
&\gamma_1 \int d\tilde{x}^4 \oint_{S_{i=1}^3} d\hat{s}^a \, \tilde{\mathcal{H}}^{a4} X + \beta_1 \gamma_1 \int d\tilde{x}^4 \oint_{S_{i=1}^3} d\hat{s}^a \, \tilde{\mathcal{H}}^{a5} X + \beta_2 \gamma_2 \int d\tilde{x}^5 \oint_{S_{i=2}^3} d\hat{s}^a \, \tilde{\mathcal{H}}^{a4} Y \\
&+ \gamma_2 \int d\tilde{x}^5 \oint_{S_{i=2}^3} d\hat{s}^a \, \tilde{\mathcal{H}}^{a5} Y + \gamma_1 \int d\tilde{x}^4 \oint_{S_\infty^3} d\hat{s}^a \, \tilde{\mathcal{H}}^{a4} X + \beta_1 \gamma_1 \int d\tilde{x}^4 \oint_{S_\infty^3} d\hat{s}^a \, \tilde{\mathcal{H}}^{a5} X \\
&+ \beta_2 \gamma_2 \int d\tilde{x}^5 \oint_{\tilde{S}_\infty^3} d\hat{s}^a \, \tilde{\mathcal{H}}^{a4} Y + \gamma_2 \int d\tilde{x}^5 \oint_{\tilde{S}_\infty^3} d\hat{s}^a \, \tilde{\mathcal{H}}^{a5} Y
\end{aligned}
$$

$$\text{(C.23)}$$

Integrating over 5-space, and so at appropriate $S_i^3$, gives

$$
\begin{aligned}
&L \left[ (\gamma_1 X_1 + \gamma_1 X_0 + \beta_2 \gamma_2 Y_0) \, Q_E^1 + (\beta_1 \gamma_1 X_1 + \beta_1 \gamma_1 X_0 + \gamma_2 Y_0) \, Q_B^1 \right] \\
&+ L \left[ (\beta_2 \gamma_2 Y_1 + \beta_2 \gamma_2 Y_0 + \gamma_1 X_0) \, Q_E^2 + (\gamma_2 Y_1 + \gamma_2 Y_0 + \beta_1 \gamma_1 X_0) Q_B^2 \right],
\end{aligned}
\qquad \text{(C.24)}
$$

with $L = \int d\tilde{x}^4 \sim \int d\tilde{x}^5$ the regulated length of both strings.

The charge associated here are: $(Q_E^1, Q_B^1) = (q_e, q_m)$; $(Q_E^2, Q_B^2) = (2q_e, q_m)$; and we re-write the above formula using the $X'$ and $Y'$ field values to obtain (4.21) in section 4.3.

## Cross term contribution

There is another non-trivial contribution to the tension formula from terms in (2.12). The term $(\zeta^1 \cdot \zeta^2)(\partial X \cdot \partial Y)$ does not vanish for the configurations for which the two string sources are non-orthogonal. Under the 5-space integral we do the integration by parts so

$$\mathcal{T}_{cr} = \int d\hat{x}_i \oint_{S_i^3} ds^a \left( \zeta^1 \cdot \zeta^2 \right) \partial_a X \, Y - \int d^5 x \, \partial^2 X \, Y. \qquad \text{(C.25)}$$

Here $d\hat{x}_i$ is the differential length element associated with the $i^{th}$ string charge core and $S_i^3$ is the hypersphere that encloses it. The second integral above which is over bulk 5d space does not contribute as the value of $\partial^2 X$ is non-zero only at the locations of the charge cores where we have defined the boundary for our integration region.

For the $(2,1) - (1,0) - (1,1)$ configuration we have two boundary regions which enclose the two string charge cores respectively, and the remaining boundaries are at the asymptotic regions near infinity. Taking account of this we can write $\mathcal{T}_{cr}$ as

$$
\begin{aligned}
\mathcal{T}_{cr} =\;& \gamma\beta \int dx^4 \oint_{S_{i=1}^3} ds^a \, \partial_a X \, Y + \gamma\beta \int d\tilde{x}^5 \oint_{S_{i=2}^3} ds^a \, \partial_a X \, Y \\
&+ \gamma\beta \int dx^4 \oint_{S_\infty^3} ds^a \, \partial_a X \, Y + \gamma\beta \int d\tilde{x}^5 \oint_{\tilde{S}_\infty^3} ds^a \, \partial_a X \, Y
\end{aligned}
\qquad \text{(C.26)}
$$

Given that the value of the $Y$ field on $\text{string}_1$ interpolate from 0 at $x^4 = 0$ to $-Y_0$ at $|x^4| \to \infty$ and the value of the $X$ field on $\text{string}_2$ from 0 at the centre to $-X_0$ far along the string, we can approximate $\mathcal{T}_{cr}$ as (for small $X_0$, $Y_0$ values)

$$\mathcal{T}_{cr} \sim -\gamma\beta \int dx^4 \oint_{S^3_{i=1}} ds^a\, \partial_a X\, Y_0 + \gamma\beta \int d\tilde{x}^5 \oint_{S^3_{i=2}} ds^a\, \partial_a X\, Y_1$$
$$- \gamma\beta \int dx^4 \oint_{S^3_\infty} ds^a\, \partial_a X\, Y_0 - \gamma\beta \int d\tilde{x}^5 \oint_{\tilde{S}^3_\infty} ds^a\, \partial_a X\, Y_0 \qquad (C.27)$$

In the asymptotic space region the third and fourth contributions vanish since the value of $X$ field is fairly constant and equal to its moduli value $-X_0$ so $\partial X \sim 0$ in this region. Thus

$$\mathcal{T}_{cr} \sim -\gamma\beta \int dx^4 \oint_{S^3_{i=1}} ds^a\, \partial_a X\, Y_0 + \gamma\beta \int d\tilde{x}^5 \oint_{S^3_{i=2}} ds^a\, \partial_a X\, Y_1 \qquad (C.28)$$

The first term is vanishingly small near marginal stability $Y_0 \sim 0$ (and the $X$ terms involve a cutoff near string-1 and so are nonsingular). For the second term, it is convenient to note that the $X$ field from $\text{string}_1$ has its largest value at the closest point between $\text{string}_1$ and $\text{string}_2$, *i.e.* at $x^4, x^5 = 0$ (elsewhere $X$ is smaller, as is $\partial X$). So we approximate

$$\partial X\Big|_{\text{string}_2} = \partial \frac{q_e}{(\vec{r} - \vec{r}_1)^2 + (x^5)^2}\Big|_{\text{string}_2} \sim \nabla_{\vec{r}} \frac{q_e}{(\vec{r} - \vec{r}_1)^2}\Big|_{\text{string}_2} \sim -q_e \frac{\vec{r}_2 - \vec{r}_1}{(\vec{r}_2 - \vec{r}_1)^4} \qquad (C.29)$$

In the limit of large transverse separation $|\vec{r}_1 - \vec{r}_2|$ between the two strings, *i.e.* $X_0, Y_0 \to 0$ near the wall of marginal stability, this term is also vanishingly small. Thus $\mathcal{T}_{cr}$ tends to zero and there is no contribution from this cross term to the mass formula in this limit.

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
