# Peer review of "M5-brane prongs, string soliton bound states and wall-crossing"

_SciPost Physics_

## Round 3 · Referee Report · Anonymous (Referee 1) · 2024-10-10

October 10, 2024

## Report on Article SciPost-2203.03674

This manuscript discusses configurations involving one dimensional M2-M5 intersections as well as interecting one dimensional configurations. The string soliton on the world volume of an M5 brane produces spike like structure which generates funnel like geometry. Since the solution is codimension 4 on the world volume of M5 brane, the scalar field profile is expected to fall off as per the inverse square law, which is consistent with what the authors find. In case of intersecting solitons also one would expect same fall off behaviour when one is sufficiently far from either of the solitons.

The manuscript is too long compared to the novelty of the results and some amount of trimming may make it a easy reading. For one, I would suggest excising appendix A completely. It is a good idea to leave out somethings to the reader to work out. Even appendix B and C can be pruned. Section 3.4 also can be reduced in size.

I also find the sentence below eq.(3.11) confusing. You have used translation invariance in $x_4$ as well as $x_5$ directions but that is not what is stated there.

There have been some works on M2-M5 system already. The one dimensional M2-M5 intersection has been worked out earlier, see e.g. `https://arxiv.org/pdf/1205.1535` which uses the PST formalism. Authors may also want to take a note of `https://arxiv.org/pdf/hep-th/9701166`.

Finally, I think the trimmed manuscript will be more suitable for SciPost Core. I will recommend publication in SciPost Core.

---

## Editorial Decision

resubmitted